# Compatible Gradient Approximations for Actor-Critic Algorithms

## Abstract

Deterministic policy gradient algorithms are foundational for actor-critic methods in controlling continuous systems, yet they often encounter inaccuracies due to their dependence on the derivative of the critic's value estimates with respect to input actions. This reliance requires precise action-value gradient computations, a task that proves challenging under function approximation. We introduce an actor-critic algorithm that bypasses the need for such precision by employing a zeroth-order approximation of the action-value gradient through two-point stochastic gradient estimation within the action space. This approach provably and effectively addresses compatibility issues inherent in deterministic policy gradient schemes. Empirical results further demonstrate that our algorithm not only matches but frequently exceeds the performance of current state-of-the-art methods by a substantial extent.

## 1  Introduction

Reinforcement learning (RL) has established itself as a prominent and effective method for addressing dynamic decision problems (Sutton and Barto, 2018). A fundamental strategy in tackling RL problems is via *policy gradient* (PG) techniques (Sutton et al., 1999), grounded in the hypothesis that actions are selected according to a parameterized distribution. PG methods iteratively update policies via stochastic gradient schemes (Sutton et al., 1999), enabling solutions in complex problems featuring continuous state-action spaces under uncertainty. PG methods are often framed within an *actor-critic* framework (Konda and Tsitsiklis, 1999), where an *actor* determines the control policy evaluated by a *critic*. Thus, the efficacy of the actor's improvements critically depends on the quality of the feedback provided by the critic. This interdependence raises nontrivial questions on the efficiency of training strategies for both the actor and the critic (D' Oro and Jaśkowski, 2020).

Conventionally, the critic is optimized through temporal difference (TD) learning (Sutton, 1988) to accurately predict the expected returns led by the actor's decisions. If the control policy is deterministic, the value estimated by the critic is solely used to compute the *action-value gradient* during policy optimization (D' Oro and Jaśkowski, 2020). This involves computing the gradient of the critic's action-value estimates with respect to the actions selected by the policy. More precisely, the *deterministic policy gradient* (DPG) (Silver et al., 2014) is determined by chaining the actor's Jacobian to the action-gradient of the critic. However, TD learning optimizes a critic that learns the *action-value*, rather than its gradient. Generally, when function approximation is used for the critic in DPG, the resulting PG may not align with the true gradient. In some cases, it may not even constitute an ascent direction (Silver et al., 2014). Nevertheless, this failure mode can be circumvented provided that $Q$-function estimates are *compatible*, meaning a set of conditions necessary for (approximately) following the true gradient; see, e.g., Theorem 3 of Silver et al. (2014).

Integration of high-dimensional scenarios requires complex function approximations, such as deep neural networks (Lillicrap et al., 2016; Fujimoto et al., 2018; Haarnoja et al., 2018). However, the use of neural networks violates the classical compatibility results, which (conveniently) calls for critics with *linear gradient representations* (Silver et al., 2014, Theorem 3). Hence, contemporary sample-based algorithms contradict the very foundations of DPG (from the angle of compatibility), thereby paving the way for potential failure scenarios (see Appendix A.2 for an illustration). Indeed, action differentiation of the $Q$-function (estimates),

i.e., action-value gradients, leaves modern successors to DPG technically unsound (D' Oro and Jaśkowski, 2020).

In this paper, we introduce (off-policy) *Compatible Policy Gradient* (oCPG), a continuous control actor-critic algorithm that approximates the action-value gradient using only value estimates from the critic. The proposed method embodies a zeroth-order nature by removing the direct computation of the action-value gradient, while provably addressing the compatibility requirement in DPG. This is achieved through evaluating critics at low-dimensional random perturbations within the action space, resulting in (batch) two-point policy gradient approximations whose distance to the true deterministic policy gradient is controllable at will.

The performance of oCPG is assessed on challenging OpenAI Gym continuous control tasks (Brockman et al., 2016), in perfect (less uncertain) and imperfect (higher uncertainty) environmental conditions. Our numerical results demonstrate that oCPG exhibits robust performance and either matches or surpasses (often substantially) the state-of-the-art in terms of stability and mean rewards. For reproducibility purposes, we share our code in the anonymous GitHub repository[1].

## 2  Related Work

Policy gradient techniques are widely used in RL. Various algorithms for estimating policy gradients have been developed; Baxter and Bartlett (2001); Metelli et al. (2018); Williams (1992) focus solely on the policy, while Mnih et al. (2016); Schulman et al. (2015; 2017); Konda and Tsitsiklis (1999); Prokhorov and Wunsch (1997) also incorporate a value function, known as actor-critic methods. Particularly, algorithms based on the deterministic policy gradient (Lillicrap et al., 2016; Silver et al., 2014) rely on the action-gradient from the critic. In function approximation, the accuracy of the critic is crucial (Barth-Maron et al., 2018); for example, applying compatibility conditions to the critic helps achieve an unbiased estimate of the policy gradient (Silver et al., 2014).

The incompatibility of the critic in DPG could be resolved by employing zeroth-order optimization (ZOO), which replaces the action-value gradient computation with an approach that does not require any (first-order) gradient information. These techniques treat the RL problem as a black-box optimization by ignoring the underlying MDP structures and directly searching for the optimal policy without gradient computations. Recent ZOO methods have proven to be competitive on common RL benchmarks (Mania et al., 2018b; Salimans et al., 2017; Lei et al., 2022), benefiting especially from not being constrained to differentiable policies. Despite these advantages, ZOO techniques suffer from high sample complexity and significant variance in gradient updates, due to their disregard for the MDP structure (Lei et al., 2022). These issues are further amplified with the increase in perturbation noise scale, potentially escalating dramatically as parameter counts can reach into the millions (Duchi et al., 2015; Lei et al., 2022). Moreover, while parameter space perturbations can induce diverse behaviors viewed as exploration, they are less beneficial in DPG methods, which are typically framed in an off-policy context (Silver et al., 2014; Lillicrap et al., 2016; Fujimoto et al., 2018). Exploration and policy learning are distinct processes in off-policy methods, making parameter-based zeroth-order techniques less effective for enhancing exploration, primarily confined to on-policy and stochastic policies (Singh et al., 2000). Contrasting with parameter space-based ZOO techniques, our approach operates in the action space and involves low-dimensional perturbations. Perturbing the action also incorporates Gaussian exploration (Fujimoto et al., 2018), which has been empirically shown to outperform parameter space exploration in off-policy settings (Saglam and Kozat, 2023). Lastly, our objective is not merely to devise a pure zeroth-order PG method but to synthesize the strengths of both ZOO and first-order approaches, employing a differentiable policy whose Jacobian is chained to the zeroth-order approximation of the action-value gradient.

In addition to the zeroth-order perspective, GProp (Balduzzi and Ghifary, 2015) and MAGE (D' Oro and Jaśkowski, 2020) are the closest known methods to our study that directly focus on optimizing and ensuring accuracy in the action-value gradient computation. GProp modifies traditional TD learning (Sutton, 1988) by using an additional neural network that predicts the action-value gradient. MAGE, drawing on recent theoretical (Saremi, 2019) and practical (Saremi and Hyvärinen, 2019) insights, trains an additional network to simulate environmental dynamics, employing it as a proxy to sample states for double gradient descent on

---

[1]https://anonymous.4open.science/r/compat-grad-approx

the $Q$-function. In stark contrast, our method (CPG) simplifies the DPG approach by directly estimating the action-value through a zeroth-order method, employing two-point evaluations of the existing critic network. This eliminates the need to train additional networks, modify TD learning frameworks, or depend on model-based settings. CPG is adaptable to a broader range of problems, bypassing the requirements for double differentiability of the $Q$-function and detailed modeling of transition dynamics. Additionally, our approach seamlessly integrates with any TD learning method and policy learning framework, and requires only a *few lines of code* to adapt to modern DPG-based algorithms.

## 3 Technical Preliminaries

We consider a standardized RL problem in which an agent moves through an environment characterized by a continuous compact state space $\mathcal{S} \subset \mathbb{R}^q$ and takes actions in a finite-dimensional action space $\mathcal{A} = \mathbb{R}^p$. Based on its action selection, the agent receives a reward from a reward function (bounded, for simplicity) $R$, where $R : \mathcal{S} \times \mathcal{A} \to \mathbb{R}$, and observes the next state $s' \in \mathcal{S}$. This generic problem is often abstracted by a Markov decision process (MDP) as a tuple $(\mathcal{S}, \mathcal{A}, P, R, \gamma)$, where $P$ denotes the environment dynamics, that is, the probabilities $P(s'|s, a)$ of moving to state $s'$ from state $s$ and if action $a$ performed, satisfying the Markov property: $P(s_{t+1}|s_i, a_i; \forall i \leq t) = P(s_{t+1}|s_t, a_t)$. The value $\gamma \in (0, 1)$ is the discount factor that prioritizes future rewards.

The action behavior of the agent is described by a policy $\pi_\theta : \mathcal{S} \to \mathcal{A}$ parameterized by $\theta \in \Theta$, which can be either a deterministic function or represent a probability distribution. Note that we may occasionally omit the parameter subscript. The agent's objective is to maximize the expected discounted sum of rewards, as quantified by the value function: $V^{\pi_\theta}(s) \coloneqq \mathbb{E}[\sum_{t=0}^{\infty} \gamma^t R(s_t, a_t \sim \pi_\theta(\cdot \mid s_t))|s_0 = s]$. By conditioning on an action, we may also define the action-value function $Q^{\pi_\theta}(s, a) \coloneqq \mathbb{E}[R(s_0, a_0) + \sum_{t=1}^{\infty} \gamma^t R(s_t, a_t \sim \pi_\theta(\cdot \mid s_t))|s_0 = s, a_0 = a]$. Lastly, let $\rho^0$ be the initial state distribution, and let $\rho^\pi$ be the $\gamma$-discounted state probability distribution induced by policy $\pi$, defined as $\rho^\pi(s') = \int_{\mathcal{S}} (1 - \gamma) \sum_{t=0}^{\infty} \gamma^t p_0(s) p(s \to s', t, \theta) \, \mathrm{d}s$, where $p(s \to s', t, \theta)$ denotes the probability density (we assume such a density exists for simplicity) at state $s'$ after starting from state $s$ and transitioning for $t$ time steps under a policy parameterized by $\theta$.

At the initial state $s$, by selecting the first action according to the policy and advancing the system, the $Q$-function becomes equivalent to the value function. Consequently, the problem is expressed as

$$\max_\theta \left\{ J(\theta) \coloneqq \mathbb{E}_{s \sim \rho^0, a \sim \pi_\theta(\cdot|s)} \left[ Q^{\pi_\theta}(s, a) \right] \right\}.$$

When the search is confined to deterministic policy parameterizations (and we do so hereafter), the objective above reduces to a simpler form, i.e.,

$$J(\theta) = \mathbb{E}_{s \sim \rho^0} \left[ Q^{\pi_\theta}(s, \pi_\theta(s)) \right]. \tag{1}$$

Under mild problem regularity conditions, it has been demonstrated by Silver et al. (2014) that the gradient of a deterministic policy may be expressed as

$$\nabla J(\theta) = \frac{1}{1 - \gamma} \int_{\mathcal{S}} \rho^{\pi_\theta}(s) \nabla_\theta \pi_\theta(s) \nabla_a Q^{\pi_\theta}(s, a) \big|_{a = \pi_\theta(s)} \, \mathrm{d}s$$

$$= \frac{1}{1 - \gamma} \mathbb{E}_{s \sim \rho^{\pi_\theta}} \left[ \nabla_\theta \pi_\theta(s) \nabla_a Q^{\pi_\theta}(s, a) \big|_{a = \pi_\theta(s)} \right], \tag{2}$$

where we use the normalizing term $1/(1 - \gamma)$ since $\rho^{\pi_\theta}$ is proper. The latter equation is known as the *deterministic policy gradient theorem* (Silver et al., 2014), which has been instrumental in the development state-of-the-art policy gradient methods essentially relying on stochastic approximation, such as Deep Deterministic Policy Gradient (DDPG) (Lillicrap et al., 2016) and Twin Delayed DDPG (TD3) (Fujimoto et al., 2018).

## 4 Off-Policy Compatible Policy Gradient

### 4.1 Problem Statement

Contemporary state-of-the-art (SOTA) first-order PG methods, such as DDPG (Lillicrap et al., 2016) and TD3 (Fujimoto et al., 2018), approximate the $Q$-function in equation 2 using a finite number of samples, which correspond to collected transition tuples. However, accurately approximating function values $Q^{\pi_\theta}$ does not necessarily imply accurate gradient approximation $\nabla_a Q^{\pi_\theta}$, and whether this happens for SOTA methods is unclear and remains an open problem. In fact, for *any* finite number of samples, it is possible to fit a function with *zero* approximation error (say, in mean-square sense) but with *arbitrarily distinct* gradients at each point, unlike the original function. This concept is illustrated with an intuitive example in Appendix A.2. In the context of PG for RL, this issue is very well-known and has been documented in detail in, e.g., (Silver et al., 2014), among several other influential papers in the literature. The hypothesis that we investigate in this work is that, if such fitting is to be employed whatsoever, then PG should be *directly* connected to the *values* of the fitted $Q$-function (using finite samples), rather than the gradients of the fitted $Q$-function.

The accuracy of the approximated PG can still be maintained with a function-approximated $Q$-function, however rather stringent conditions have to be met (Silver et al., 2014, Theorem 3). For completeness, we provide a detailed summary of these compatibility requirements in Appendix A.1. A critical point in Silver et al. (2014) for ensuring such compatibility is the convenient assumption that the gradient of the $Q$-function is linear in its parameters, i.e., $\nabla_a Q_\omega(s,a)|_{a=\pi_\theta(s)} = \nabla_\theta \pi_\theta(s)^\top \omega$, where $\omega$ denotes the parameters. However, it has been demonstrated that scaling to large and complex state-action spaces inevitably requires the use of more expressive representations, such as deep neural networks (Lillicrap et al., 2016). This necessity potentially compromises compatibility and results in $Q$-function gradients that may not accurately follow the true policy gradient (Silver et al., 2014).

This failure mode has recently been linked to the need to minimize the norm of the action-value gradient of the policy evaluation error, rather than its value, in order to reduce the approximation error of the computed PG (D' Oro and Jaśkowski, 2020, Proposition 3.1). Aligning with this insight, double backpropagation through the TD error may address this issue in scenarios where model-based approaches are feasible (D' Oro and Jaśkowski, 2020). However, model-free RL is often preferred due to its effectiveness in environments with complex or unknown dynamics, offering enhanced flexibility and robustness for direct policy learning via interactions with the environment (Sutton and Barto, 2018); this is the path taken herein.

While several studies have pursued the goal of overcoming the incompatibility of the critic (D' Oro and Jaśkowski, 2020; Balduzzi and Ghifary, 2015), our goal herein is to consistently approximate the deterministic policy gradient $\nabla J$ as

$$\hat{\nabla} J(\theta) = \frac{1}{1-\gamma} \mathbb{E}_{s \sim \rho^\pi} \left[ \nabla_\theta \pi_\theta(s) \hat{\nabla}_a Q^{\pi_\theta}(s,a)) \big|_{a=\pi_\theta(s)} \right], \tag{3}$$

where $\hat{\nabla}_a Q^{\pi_\theta}(s,a)$ represents an appropriately designed approximation of $\nabla_a Q^{\pi_\theta}(s,a)$, resulting in a $\hat{\nabla} J$ that is *provably compatible*, *general* and *implementable in a model-free manner*. To the best of our knowledge, no study has yet proposed a gradient surrogate (i.e., $\hat{\nabla} J$) that possesses all aforementioned features *simultaneously*, in a model-free deep RL setting. As we empirically demonstrate later (Section 5), such an approach in fact results in substantial operational benefits over the state-of-the-art.

### 4.2 Provably Compatible Policy Gradient Approximations

To begin, we temporarily assume that $Q^{\pi_\theta}$ is available and, for a *smoothing parameter* $\mu > 0$, we consider the $\mu$-smoothed Q-function defined as

$$Q_\mu^{\pi_\theta}(s,a) \coloneqq \mathbb{E}_{\boldsymbol{u}}\left[Q^{\pi_\theta}(s, a + \mu\boldsymbol{u})\right], \ \boldsymbol{u} \sim \mathcal{N}(0, I_p), \tag{4}$$

provided that the smoothing is well-defined. Smoothing may be equivalently thought of as enforcing exploration, and the random perturbation $\mu\boldsymbol{u}$ may be thought of as the purely stochastic part of a standard *Gaussian policy* $\pi_\theta^\mu(\cdot \mid s) \coloneqq \pi_\theta(s) + \mu\boldsymbol{u}$, with $\boldsymbol{u} \sim \mathcal{N}(0, I_p)$. Here, it is helpful to view such a Gaussian policy as a deterministic policy perturbed by noise in the action space, which is usually of substantially

lower dimension as compared with the dimension of the corresponding parameter space (Mania et al., 2018a; Maheswaranathan et al., 2019; Faccio et al., 2021; Müller et al., 2021).

Implementing (Gaussian) smoothing on deterministic actions aligns with approaches used in seminal works such as by Lillicrap et al. (2016); Fujimoto et al. (2018). As mentioned above, this technique introduces stochasticity in action selection, which prevents premature convergence to suboptimal policies by promoting exploration of a broader range of state-action pairs (Sutton and Barto, 2018; Fujimoto et al., 2018). In any case, though, the ultimate goal is to discover a near-optimal *deterministic* control policy that near-solves equation 1. In our context, we achieve this by learning a deterministic policy through trajectories generated by a smoothed/perturbed policy, with all performance assessments based on deterministic actions.

Our development will be relying on the following basic result by Nesterov and Spokoiny (2017), which is central in the development and analysis of zeroth-order optimization methods, and establishes that the gradient of smoothed functions such as that in equation 4 may be evaluated in a model-free fashion, through batches of two-point evaluations of the function itself.

**Proposition 1** (Nesterov and Spokoiny 2017). *Let $f : \mathbb{R}^p \to \mathbb{R}$ be a bounded function. For every $\mu > 0$, the smoothed function $f_\mu(x) := \mathbb{E}_{\boldsymbol{u}}\big[f(x + \mu\boldsymbol{u})\big]$, $\boldsymbol{u} \sim \mathcal{N}(0, I_p)$ is well-defined, differentiable and its gradient admits the representation*

$$\nabla f_\mu(x) = \mathbb{E}_{\boldsymbol{u} \sim \mathcal{N}(0, I_p)}\left[\frac{f(x + \mu\boldsymbol{u}) - f(x)}{\mu}\boldsymbol{u}\right], \ \forall x \in \mathbb{R}^p.$$

*Further, if $f$ is $G$-smooth (i.e., with $G$-Lipschitz gradients), it holds that*

$$\sup_x \|\nabla f_\mu(x) - \nabla f(x)\| \le \mu G \sqrt{p}.$$

Proposition 1 easily motivates the $Q$-function *zeroth-order gradient approximation*

$$\nabla_a Q_\mu^{\pi_\theta}(s, a) = \mathbb{E}_{\boldsymbol{u}}\left[\frac{Q^{\pi_\theta}(s, a + \mu\boldsymbol{u}) - Q^{\pi_\theta}(s, a)}{\mu}\boldsymbol{u}\right],$$

resulting in a deterministic policy gradient approximation reading (cf. 3)

$$\hat{\nabla}^\mu J(\theta) := \frac{1}{1 - \gamma}\mathbb{E}_{s \sim \rho^{\pi_\theta}, \boldsymbol{u} \sim \mathcal{N}(0, I_p)}\left[\nabla_\theta \pi_\theta(s) \times \frac{Q^{\pi_\theta}(s, a + \mu\boldsymbol{u}) - Q^{\pi_\theta}(s, a)}{\mu}\boldsymbol{u}\bigg|_{a = \pi_\theta(s)}\right].$$

While $\hat{\nabla}^\mu J$ is a genuine and consistent model-free approximation of the deterministic policy gradient $\nabla J$ (through Proposition 1), it requires access to (evaluations of) the $Q$-function itself, the latter being both unknown (in the majority of cases) and also hard, or at least non-trivial to sample.

**Approximating the $Q$-function** Following the actor-critic paradigm (Konda and Tsitsiklis, 1999), we approximate the $Q$-function by a parameterized learning representation $Q_\psi$ (the critic), for instance, a deep neural network (i.e., $Q$-network), similarly to the control policy $\pi_\theta$ (or actor). We denote $Q$-network parameters abstractly with the subscript $\psi$ (different than the smoothing parameter $\mu$). Then, we propose the parameterized deterministic policy gradient approximation

$$\hat{\nabla}^{\mu,\psi} J(\theta) := \frac{1}{1 - \gamma}\mathbb{E}_{s \sim \rho^{\pi_\theta}, \boldsymbol{u}}\left[\nabla_\theta \pi_\theta(s) \times \frac{Q_\psi(s, a + \mu\boldsymbol{u}) - Q_\psi(s, a)}{\mu}\boldsymbol{u}\bigg|_{a = \pi_\theta(s)}\right]. \tag{5}$$

Given equation 5, a natural question is how the representation $Q_\psi$ should be trained, in order for $\hat{\nabla}^{\mu,\psi} J$ to achieve a small approximation error as compared with the true policy gradient $\nabla J$. In other words, instead of approximating in value, we would like to choose $\psi$ such that the representation $Q_\psi$ *is (approximately) compatible to $Q^{\pi_\theta}$* (in the standard sense of Silver et al.). To this end, let us first define the *perturbation representation error* (PRE):

$$\varepsilon_{\mu,\psi}^{\pi_\theta} := \sqrt{\mathbb{E}_{s, \boldsymbol{u}}\big[\big|Q_\psi(s, \pi_\theta(s) + \mu\boldsymbol{u}) - Q^{\pi_\theta}(s, \pi_\theta(s) + \mu\boldsymbol{u})\big|^2\big]},$$

where $\theta, \mu$ and $\psi$ are given. We have the following result (see Appendix A.3 for the proof).

**Theorem 1** ($\epsilon$-**Compatible Policy Gradient**). *Let $B > 0$ satisfy $\sup_{(s,\theta) \in \mathcal{S} \times \Theta} \|\nabla_\theta \pi_\theta(s)\| \leq B$. Then the gradient approximation $\hat{\nabla}^{\mu,\psi} J$ satisfies, for every $\theta \in \Theta, \mu > 0$ and $\psi$,*

$$\|\hat{\nabla}^{\mu,\psi} J(\theta) - \nabla J(\theta)\| \leq \frac{B}{1-\gamma} \left[ \frac{\varepsilon^{\pi_\theta}_{\mu,\psi}}{\mu} \sqrt{p} + \mathbb{E}_s \left[ \left\| \nabla_a Q^{\pi_\theta}_\mu(s,a) - \nabla_a Q^{\pi_\theta}(s,a) \right\| \big|_{a=\pi_\theta(s)} \right] \right].$$

*In particular, if $Q^{\pi_\theta}(s,\cdot)$ is itself $G$-smooth (uniformly), it follows that, for every $\theta \in \Theta$,*

$$\|\hat{\nabla}^{\mu,\psi} J(\theta) - \nabla J(\theta)\| \leq \frac{B\sqrt{p}}{1-\gamma} \left[ \frac{\varepsilon^{\pi_\theta}_{\mu,\psi}}{\mu} + G\mu \right].$$

The usefulness of the stated result is twofold: On one hand, Theorem 1 quantifies the performance of $\hat{\nabla}^{\mu,\psi} J$ in terms of smoothing parameter ($\mu$) and the PRE ($\psi$). On the other hand, Theorem 1 reveals *how* $\psi$ can be chosen such that the gradient approximation error achieved by adopting $\hat{\nabla}^{\mu,\psi} J$ is controlled *at will*, achieving the desirable policy gradient alignment. To illustrate this, consider a smooth $Q$-function for clarity. Theorem 1 then implies that

$$\inf_\psi \|\hat{\nabla}^{\mu,\psi} J(\theta) - \nabla J(\theta)\| \leq \frac{B\sqrt{p}}{1-\gamma} \left[ \frac{\inf_\psi \varepsilon^{\pi_\theta}_{\mu,\psi}}{\mu} + G\mu \right]. \tag{6}$$

We first observe that the smoothing term $\mu$ appears in both the numerator and denominator of the upper bound. For any fixed $\mu > 0$ (and at each actor instance indexed by $\theta$), one can *optimally train* the critic $Q_\psi$ to minimize the perturbation representation error $\varepsilon^{\pi_\theta}_{\mu,\psi}$. With a sufficiently expressive function class, it is theoretically possible to achieve an *arbitrarily small* gradient approximation error. In the ideal case, one could overfit the critic at each gradient evaluation step, eliminating the first term on the right-hand side of equation 6 and reducing the total gradient approximation error to $\mathcal{O}(\mu)$.

In practice, however, this term does not vanish entirely. A positive $\mu > 0$ must balance compatibility and exploration by acting as a smoothing factor. Achieving this balance requires sufficiently accurate $Q$-value estimation—in the sense of minimizing $\varepsilon^{\pi_\theta}_{\mu,\psi}$—to preserve gradient fidelity under a fixed $\mu$, which also controls the degree of Gaussian exploration (Fujimoto et al., 2018). In any case, minimizing the PRE with respect to $\psi$ is a well-justified strategy for ensuring policy gradient compatibility, as supported by Theorem 1. By contrast, standard DPG methods may suffer from uncontrolled gradient error due to potential incompatibility (see Section 4.1).

Motivated by these observations, we refer to $\hat{\nabla}^{\mu,\psi} J$ as a *($\epsilon$-)Compatible Policy Gradient* (CPG). Further discussion on our notion of (approximate) $\epsilon$-compatibility intruduced herein and its generality when compared with standard (exact) compatibility as originally formulated in Silver et al. (2014) is presented in Appendix A.4.

### 4.2.1 Further Remarks

*First*, we note that Theorem 1 is general in scope—it does not assume any specific state distribution (see also the proof in the supplement). The use of the on-policy distribution $\rho^{\pi_\theta}$ in this section is purely a narrative choice. The arguments leading up to, and including, Theorem 1 apply to any state distribution, including empirical ones—such as those induced from the replay buffer in off-policy settings (see Section 4.3). The use of the notation $\mathbb{E}_s[\cdot]$ in Theorem 1 is deliberate, highlighting that no particular sampling distribution is assumed.

*Second*, the reader may notice that the term $Q_\psi(s,a)\boldsymbol{u}|_{a=\pi_\theta(s)}$ in equation 5 has mean zero and could be omitted. This is also true in Proposition 1, and indeed, Theorem 1 holds verbatim for the resulting one-sided gradient surrogate. We retain this term in equation 5 to ensure finite variance under the expectation $\mathbb{E}_{s,\boldsymbol{u}}$, where it functions analogously to a baseline. In practice, this additional term improves the stability of the gradient approximation—often substantially.

*Third*, convergence guarantees can be obtained using standard techniques in settings such as on-policy actor-critic schemes, where the critic is fully optimized before or after each actor update. While this direction is beyond the scope of our current work, it offers promising avenues for future research.

*Lastly*, while the Gaussian exploration noise appears in the definition of $\varepsilon^{\pi_\theta}_{\mu,\psi}$, the objective remains deterministic policy optimization—as in off-policy methods like DDPG (Lillicrap et al., 2016, Algorithm 1, lines 8–13) and TD3 (Fujimoto et al., 2018, Algorithm 1, lines 6–11). However, both methods use gradients of fitted $Q$-functions, which may violate compatibility, as discussed in Section 4.1. Theorem 1 suggests that equation 5 can be used instead as a provably compatible alternative for gradient approximation.

### 4.3   Off-Policy Deep Reinforcement Learning

To devise an efficient learning framework centered on CPG, we incorporate standard components from existing literature.

**Clipped Double $Q$-learning (Fujimoto et al., 2018)**   Based on our discussion above, to exploit $\hat{\nabla}^{\mu,\psi}J$ as a gradient surrogate (and thus harness the benefits of Theorem 1), we should learn a $Q_{\psi^*}$ such that $\psi^* \in \arg\min_\psi \varepsilon^{\pi_\theta}_{\mu,\psi}$ (iteration-wise for fixed $\theta$ and $\mu$). However, learning such a $Q_{\psi^*}$ assumes the feasibility of obtaining unbiased estimates of the $Q$-function at any state-action (perturbation) pair, as well as sampling from the discounted state distribution $\rho^\pi$. Specifically for obtaining unbiased $Q$-value estimates in the context of deep RL, one can employ the practical clipped double $Q$-learning algorithm (Fujimoto et al., 2018) to learn the $Q$-network (in conjunction with an experience replay buffer – see next paragraph), which has been shown to eliminate the estimation bias effectively.

**Off-Policy Learning**   Our selection of an off-policy learning approach is driven by its potential for high sample efficiency, notably through the use of an experience replay buffer (Lin, 1992). The agent stores samples as transition tuples $(s, a, R(s,a), s')$ in the buffer and periodically samples a minibatch of these transitions for policy and $Q$-network updates. Building upon this, CPG, although theoretically on-policy, is implemented in an off-policy framework. This approach, however, inherently involves a distributional shift, as the samples used may significantly diverge from the current agent's policy (Sutton and Barto, 2018). Despite this, our method draws on the strategy employed by modern off-policy algorithms (Silver et al., 2014; Lillicrap et al., 2016; Fujimoto et al., 2018; Haarnoja et al., 2018), which simplifies the complexities by approximating off-policy samples as on-policy.

Combining the components introduced, we present our proposed CPG method in an off-policy deep RL setting, implementing stochastic gradient ascent via the implementable gradient approximation

$$\hat{\nabla}^{\mu,\psi}J(\theta) \approx \hat{\nabla}^{\mu,\psi}_\beta J(\theta) = \frac{1}{1-\gamma}\mathbb{E}_{s\sim\rho^\beta,\boldsymbol{u}}\left[\nabla_\theta\pi_\theta(s)\frac{Q_\psi(s, a+\mu\boldsymbol{u}) - Q_\psi(s,a)}{\mu}\boldsymbol{u}\bigg|_{a=\pi_\theta(s)}\right],$$

where $\beta$ denotes the set of policies that collected the off-policy samples (referring to the replay buffer), which can also be on-policy, and $\rho^\beta$ represents the corresponding (approximation to the) state visitation distribution. Recognizing the limitations of this approximation, we focus on the immediate performance of our algorithm. We note that a detailed analysis of the "off-policyness" (see, e.g., Munos et al.; Saglam et al.), as well as the complexity and convergence analysis of this actor-critic setup under CPG, are not the topics of our current scope, and thus deferred for future research.

The pseudocode for the off-policy CPG (oCPG) is provided in Algorithm 1. The inclusion of dual $Q$-networks, target parameters, and delayed policy updates draws from the structure of Clipped Double $Q$-learning. We also consistently optimize the policy with respect to the first $Q$-network to ensure stability.

## 5   Experiments

We benchmark the performance of oCPG against state-of-the-art PG-based actor-critic algorithms: TD3 (Fujimoto et al., 2018), Soft Actor-Critic (SAC) (Haarnoja et al., 2018), and Proximal Policy Optimization (PPO). Since TD3 is a variant of DPG, comparing it with CPG provides a strong test of our proposed PG method against traditional DPG in deep RL. Likewise, as SAC extends the standard stochastic PG algorithm, comparisons with SAC highlight CPG's effectiveness relative to stochastic PG in the deep RL setting.

---

**Algorithm 1** Off-Policy Compatible Policy Gradient (oCPG)

---
1: Initialize the policy network $\pi_\theta$, and critic networks $Q_{\psi_1}$, $Q_{\psi_2}$ with random parameters $\theta$, $\psi_1$, $\psi_2$
2: Initialize target networks $\theta' \leftarrow \theta$, $\psi_1' \leftarrow \psi_1$, $\psi_2' \leftarrow \psi_2$
3: Initialize the experience replay buffer: $\mathcal{B} = \emptyset$
4: **for** $t = 1$ to $T$ **do**
5:      Select action with Gaussian exploration noise $a = \pi_\theta(s) + \mu \boldsymbol{u}$, where $\boldsymbol{u} \sim \mathcal{N}(0, I_p)$
6:      Execute action $a$, receive reward $r = R(s, a)$, and observe new state $s'$
7:      Store the collected transition tuple in the replay buffer: $\mathcal{B} \leftarrow \mathcal{B} \cup (s, a, r, s')$
8:      Sample a minibatch of $N$ transitions from the replay buffer: $(\mathbf{s}, \mathbf{a}, \mathbf{r}, \mathbf{s}') \sim \mathcal{B}$
9:      $\tilde{\mathbf{a}} \leftarrow \pi_{\theta'}(\mathbf{s}') + \epsilon$, where $\{\epsilon_i \sim \text{clip}\left(\mathcal{N}(0, \sigma^2 I_p), -c, c\right)\}_{i=1}^N$      ▷ Clipped Double $Q$-learning
10:      $\mathbf{y} \leftarrow \mathbf{r} + \gamma \min_{i=1,2} Q_{\psi_i'}(\mathbf{s}', \tilde{\mathbf{a}})$      ▷ Clipped Double $Q$-learning
11:      Update critics: $\psi_j \leftarrow \text{argmin}_{\psi_j} \frac{1}{N} \sum_i \left(\mathbf{y}_i - Q_{\psi_j}(\mathbf{s}_i, \mathbf{a}_i)\right)^2$      ▷ Clipped Double $Q$-learning
12:      **if** $t \bmod d$ **then**      ▷ Delayed policy updates
13:          Update the policy by CPG:

$$\hat{\nabla}_\beta^{\mu, \psi} J(\theta) = \frac{1}{N} \sum_i \nabla_\theta \pi_\theta(\mathbf{s}_i) \frac{Q_{\psi_1}(\mathbf{s}_i, \mathbf{a}_i + \mu \boldsymbol{u}_i) - Q_{\psi_1}(\mathbf{s}_i, \mathbf{a}_i)}{\mu} \boldsymbol{u}_i \Bigg|_{\substack{\mathbf{a}_i = \pi_\theta(\mathbf{s}_i) \\ \boldsymbol{u}_i \sim \mathcal{N}(0, I_p)}}$$

14:          Update target networks:

$$\theta' \leftarrow \tau \cdot \theta + (1 - \tau) \cdot \theta'$$
$$\psi_i' \leftarrow \tau \cdot \psi_i + (1 - \tau) \cdot \psi_i'$$

15:      **end if**
16: **end for**

---

## 5.1 Experimental Setup

**Evaluation**    Performance is assessed on continuous control tasks from the MuJoCo suite (Todorov et al., 2012) using OpenAI Gym (Brockman et al., 2016). Each algorithm is trained until convergence in evaluation reward, with evaluations performed every 1,000 steps in a noise-free environment using deterministic actions. Results are averaged over 10 random seeds, capturing variability from the Gym simulator, network initialization, and stochastic processes.

**Implementation and Hyperparameters**    For simplicity, the implementation of CPG omits the $1 - \gamma$ term in the denominator. To ensure a fair comparison with TD3 and isolate the impact of our proposed PG method, we used the same hyperparameters (e.g., learning rate), with the only difference being how the policy gradient is computed.

**Selection of** $\mu$    In Section 4.2, we highlighted the importance of selecting the smoothing term $\mu$. Initial simulations tested $\mu = \{0.025, 0.05, 0.075, 0.1, 0.125\}$. While 0.1 and 0.125 performed similarly, results worsened as $\mu$ decreased. This reflects the nuanced balanced between $\mu$ and the $Q$-function approximation error $\varepsilon_{\mu, \psi}^{\pi_\theta}$ in the PG error's upper bound, i.e., equation 6. It appears that the $Q$-function approximation could not sufficiently compensate for $\mu < 0.1$, suggesting a more expressive $Q$-function approximation. To choose between 0.1 and 0.125, we aligned $\mu$ with the standard deviation of the Gaussian exploration noise used in TD3, which is 0.1. This alignment ensures *(i)* fair evaluations with TD3, *(ii)* an adequate exploration, and *(iii)* a moderate value for smoothing. Notably, our method does not introduce any additional hyperparameters, as compared to the current state-of-the-art.

A detailed description of the experimental setup and implementation can be found in Appendix B.2.

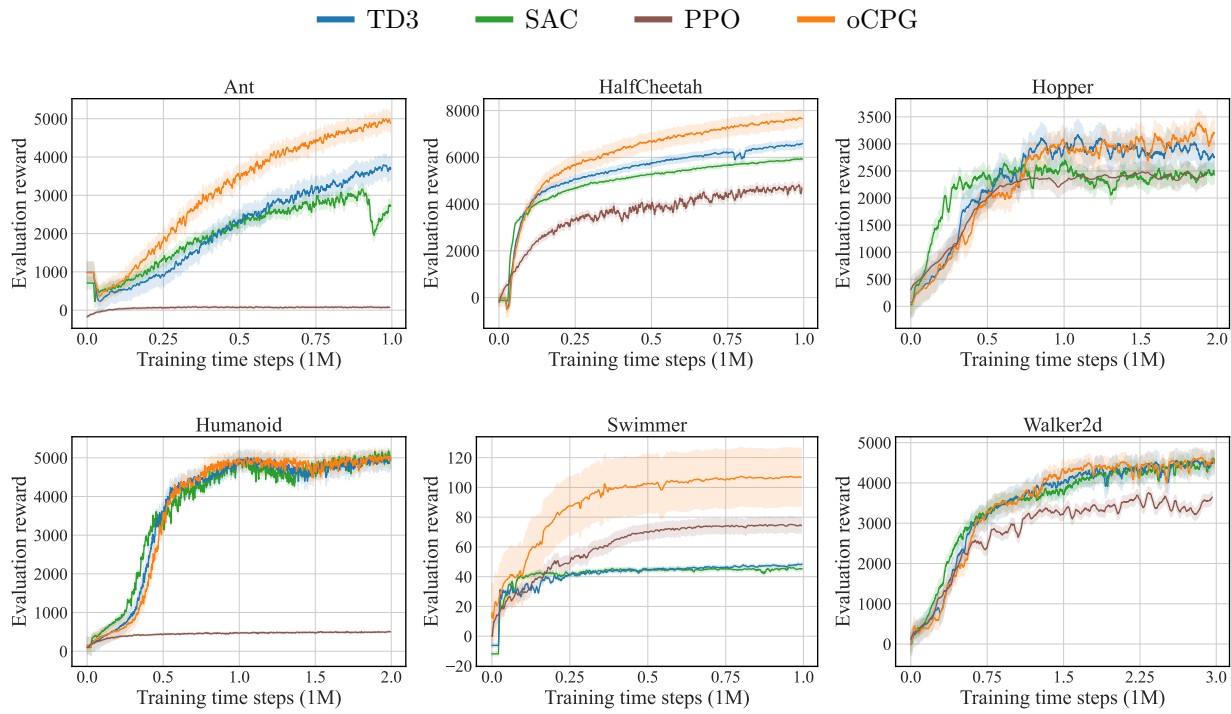

Figure 1: Learning curves for benchmark MuJoCo environments, averaged over 10 random seeds. Evaluation reward is calculated as the undiscounted sum of rewards collected by the agent during each evaluation episode. The shaded region indicates the 95% confidence interval of the mean performance.

## 5.2 Results

The results are presented in Figure 1. Additionally, Table 1 reports the converged rewards, calculated as the mean of the last 50 evaluation rewards, along with several statistical tests to assess the significance of performance differences. We examine these statistical tests in Appendix C.1, highlighting their computational differences and underlying assumptions.

In most environments, oCPG achieves faster convergence to higher final rewards. Although the performance curves are intertwined, making it challenging to visually identify a clear leader, the four statistical tests reported in Table 1 confirm that oCPG is the best-performing algorithm in three out of six environments. In the remaining environments, it is on par with the baselines.

Notice that the only difference between oCPG and TD3 lies in how the PG is computed. All other configurations, including the degree of exploration $\mu$ (which also influences the smoothness in our PG algorithm), remain unchanged. Thus, the statistically significant performance improvement can be attributed to the PG computation. While we do not explicitly claim it, this observation suggests that CPG may produce more accurate PG estimates than DPG.

We observe that oCPG converges more slowly than TD3 in the Humanoid environment, where the combined state-action dimension is 393. In our experiments, the $Q$-networks have only 256 hidden units per layer, and neural networks are sensitive to input noise, which becomes more noticeable as the input size surpasses the network's capacity. This issue arises due to the smoothing step on line 13 in Algorithm 1; however, using larger networks could mitigate this limitation. Despite this, oCPG achieves the highest converged rewards in five of six environments. While its performance in the Swimmer task is statistically significant, the rewards show high variance because some seeds converge to suboptimal levels, leading to wide confidence intervals.

Table 1: Mean of the last 50 evaluation rewards on benchmark MuJoCo environments, averaged over 10 random seeds, with $\pm$ denoting a 95% confidence interval. The highest performance is marked in **boldface**. The statistically superior algorithm is highlighted (where applicable), while the highest performances shared by multiple algorithms are indicated. Welch's test, Student's t-test, paired t-test, and the Wilcoxon rank-sum test are conducted at a significance level of 0.05, all yielding consistent results across methods.

| Environment | TD3 | SAC | PPO | oCPG |
|---|---|---|---|---|
| Ant | $4225.080 \pm 835.236$ | $3945.508 \pm 673.329$ | $221.918 \pm 136.231$ | **5284.923 $\pm$ 412.932** |
| HalfCheetah | $6744.592 \pm 671.739$ | $6074.230 \pm 405.968$ | $5089.677 \pm 2058.217$ | **7956.502 $\pm$ 1178.024** |
| Hopper | $3620.488 \pm 137.280$ | $3585.355 \pm 186.848$ | $2785.950 \pm 1433.114$ | **3687.673 $\pm$ 118.093** |
| Humanoid | $5289.623 \pm 148.741$ | **5567.597 $\pm$ 141.133** | $582.319 \pm 79.251$ | $5348.386 \pm 109.030$ |
| Swimmer | $59.806 \pm 9.477$ | $58.396 \pm 10.990$ | $76.471 \pm 9.124$ | **123.130 $\pm$ 59.253** |
| Walker2d | $4842.336 \pm 723.696$ | $4337.161 \pm 2313.581$ | $4725.314 \pm 492.717$ | **4953.588 $\pm$ 298.321** |

Moreover, discrepancies in reported reward levels compared to the TD3 and SAC papers may result from differences in environmental stochasticity and the MuJoCo simulator version. Nevertheless, consistent performance across multiple random seeds provides a reliable basis for evaluating our method. This approach aligns with established deep RL experimentation standards (Henderson et al., 2018), which prioritize comparative analysis over absolute performance.

Finally, the smoothing parameter $\mu$ is linked to exploration levels, eliminating the need for environment-specific tuning. Our experiments show that a fixed value of $\mu = 0.1$ generalizes well across diverse environments, ensuring effective policy gradient approximation while preserving exploratory behavior.

**Imperfect Environmental Conditions**  To evaluate the performance of our method *under uncertainty*, we present results with perturbed rewards in Appendix C. Convergence is noticeably affected in environments with altered rewards, as uncertainty distorts the agent's received rewards, directly influencing the $Q$-value used by CPG for policy gradient computation. Although the baselines also depend on $Q$-value estimates for their gradients, reward modifications have minimal effect on their final performance. This demonstrates that the precision of value gradient estimates does not necessarily correlate with the accuracy of the value estimates themselves, as discussed in Appendix A.2. Despite these challenges, oCPG achieves higher reward levels over time.

## 6    Conclusion and Future Work

We introduced Compatible Policy Gradient (CPG), a policy gradient technique for actor-critic algorithms that employs a zeroth-order approximation method to estimate the action-value gradient in deterministic policy gradient (DPG) algorithms. By employing stochastic two-point gradient estimation with low-dimensional perturbations in action space, our approach eliminates the need for explicit action-value gradient computation, addressing compatibility issues commonly faced in traditional DPG methods with deep function approximation. Empirical results on various continuous control tasks demonstrate that CPG outperforms state-of-the-art methods.

Our findings highlight two key implications. First, they reveal the potential of zeroth-order (gradient-free) methods to overcome challenges related to accurate derivative computations in policy optimization. Second, the strong performance of CPG in complex continuous control environments suggests that similar approaches could prove effective in other reinforcement learning domains characterized by uncertain dynamics or imperfect system models.

**Future Work**  An off-policy correction mechanism for CPG could be developed by leveraging the probability ratio between the agent's current policy and its behavioral policy. Additionally, while we did not investigate

convergence results for on-policy actor-critic methods with fully optimized critics that achieve near-zero approximation error, this remains a compelling direction for future research.

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

## A  Supporting Details for Section 4

### A.1  Compatibility Requirements in Deterministic Policy Gradients

Here, for completeness we recall the classical result of Silver et al. (2014) (specifically Theorem 3 of Silver et al.), which ensures *exact gradient compatibility*, however at the expense of restrictive conditions on the form of the adopted $Q$-function approximation (in particular, its gradient must be of linear form).

**Theorem 2** (**Compatible Function Approximation**). *A function approximator $Q_\omega(s, a)$ is compatible with a deterministic policy $\pi_\theta(s)$, $\nabla J(\theta) = \mathbb{E}_{s \sim \rho^{\pi_\theta}} \left[ \nabla_\theta \pi_\theta(s) \nabla_a Q_\omega(s, a) \big|_{a = \pi_\theta(s)} \right]$, if*

1. *$\nabla_a Q_\omega(s, a) \big|_{a = \pi_\theta(s)} = \nabla_\theta \pi_\theta(s)^\top \omega$ and*

2. *$\omega$ minimizes the **gradient** mean-squared error $\mathbb{E}_s \left[ \left\| \nabla_a Q_\omega(s, a) \big|_{a = \pi_\theta(s)} - \nabla_a Q^{\pi_\theta}(s, a) \big|_{a = \pi_\theta(s)} \right\|^2 \right]$.*

### A.2 Concept Illustration: Accurate Value Approximation does *not* Guarantee Accurate Gradient Approximation

We consider the function fitting problem (also sometimes called Runge's phenomenon) illustrated in Figure 2. As shown in the figure, two distinct functions are fitted to a finite set of sample points, each achieving *zero* approximation error—e.g., in the mean-square sense—since they align exactly with the true function at those points. However, given the finite nature of the samples, there exist *infinitely many* such interpolants. While the approximation error is identically zero, their gradients (i.e., slopes) can differ significantly from one another and from the true function, leading to arbitrarily different gradient directions—particularly at the sample locations.

This highlights a fundamental issue: even as function approximations improve, gradient estimates may remain inconsistent due to the inherent ambiguity in interpolating between a finite set of samples. Each interpolation yields a different gradient profile, regardless of approximation accuracy. Our randomized smoothing approach, grounded in Proposition 1, directly addresses this issue by mitigating gradient inconsistency.

This observation has direct implications for sample-based RL methods. No matter the size of the replay buffer, the $Q$-function is always trained on a finite dataset. As a result, gradient inconsistencies are inevitable when computing the policy gradient using the (generally incompatible) gradients of the fitted $Q$-function.

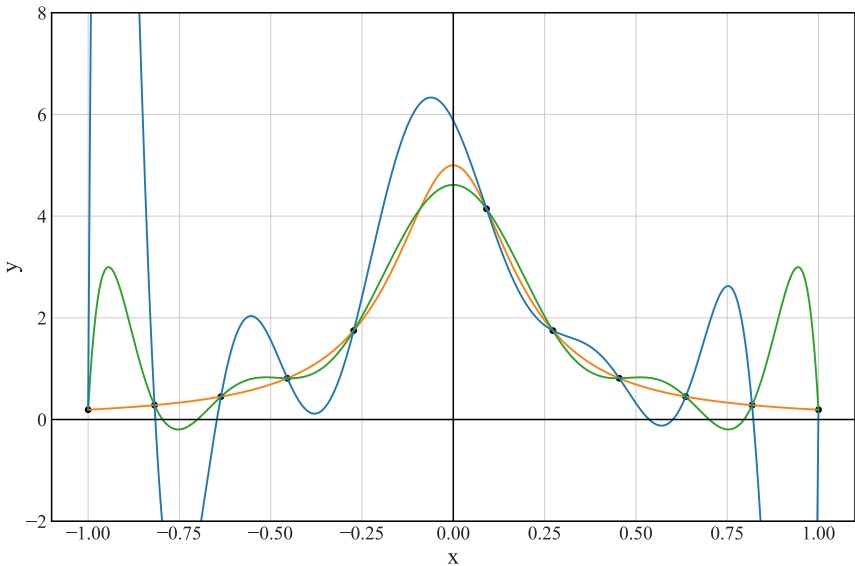

Figure 2: Sketch of a function fitting problem (Runge's phenomenon) containing finite number of samples.

## A.3  Proof of Theorem 1

We may write, for every $\theta \in \Theta, \psi, \mu > 0$,

$$
\begin{aligned}
&(1-\gamma)\|\hat{\nabla}J(\theta) - \nabla J(\theta)\| \\
&= \left\| \mathbb{E}_{s,\boldsymbol{u}} \left[ \nabla_\theta \pi_\theta(s) \frac{Q_\psi(s, a+\mu\boldsymbol{u}) - Q_\psi(s,a)}{\mu} \boldsymbol{u} \right] \Big|_{a=\pi_\theta(s)} - \mathbb{E}_s \left[ \nabla_\theta \pi_\theta(s) \nabla_a Q^{\pi_\theta}(s,a) \right] \Big|_{a=\pi_\theta(s)} \right\| \\
&= \left\| \mathbb{E}_s \left[ \nabla_\theta \pi_\theta(s) \mathbb{E}_{\boldsymbol{u}} \left[ \frac{Q_\psi(s, a+\mu\boldsymbol{u}) - Q_\psi(s,a)}{\mu} \boldsymbol{u} - \nabla_a Q^{\pi_\theta}(s,a) \right] \Big|_{a=\pi_\theta(s)} \right] \right\| \\
&\leq \mathbb{E}_s \left[ \left\| \nabla_\theta \pi_\theta(s) \mathbb{E}_{\boldsymbol{u}} \left[ \frac{Q_\psi(s, a+\mu\boldsymbol{u}) - Q_\psi(s,a)}{\mu} \boldsymbol{u} - \nabla_a Q^{\pi_\theta}(s,a) \right] \Big|_{a=\pi_\theta(s)} \right\| \right] \\
&\leq B \mathbb{E}_s \left[ \left\| \mathbb{E}_{\boldsymbol{u}} \left[ \frac{Q_\psi(s, a+\mu\boldsymbol{u}) - Q_\psi(s,a)}{\mu} \boldsymbol{u} - \nabla_a Q^{\pi_\theta}(s,a) \right] \right\| \Big|_{a=\pi_\theta(s)} \right],
\end{aligned}
$$

where we have used Jensen (norms are convex), (matrix) Cauchy-Schwarz, and the assumption that $\sup_{s,\theta} \|\nabla_\theta \pi_\theta(s)\| \leq B$. In passing, we note that a non-standard (i.e., transposed) definition for the Jacobian $\nabla_\theta \pi_\theta(s)$ was used when applying the chain rule. Additionally, the latter expression remains valid as $\mu \to 0$ in which case we obtain the *deterministic policy gradient error* reading

$$
(1-\gamma)\|\hat{\nabla}J(\theta) - \nabla J(\theta)\| \leq B \mathbb{E}_s \left[ \left\| \nabla_a Q_\psi(s,a) - \nabla_a Q^{\pi_\theta}(s,a) \right\| \Big|_{a=\pi_\theta(s)} \right].
$$

We further have, for every $(s,a) \in \mathcal{S} \times \mathcal{A}$,

$$
\begin{aligned}
&\left\| \mathbb{E}_{\boldsymbol{u}} \left[ \frac{Q_\psi(s, a+\mu\boldsymbol{u}) - Q_\psi(s,a)}{\mu} \boldsymbol{u} \right] - \nabla_a Q^{\pi_\theta}(s,a) \right\| \\
&= \left\| \mathbb{E}_{\boldsymbol{u}} \left[ \frac{Q_\psi(s, a+\mu\boldsymbol{u}) - Q_\psi(s,a)}{\mu} \boldsymbol{u} \right] - \nabla_a Q_\mu^{\pi_\theta}(s,a) + \nabla_a Q_\mu^{\pi_\theta}(s,a) - \nabla_a Q^{\pi_\theta}(s,a) \right\| \\
&= \left\| \mathbb{E}_{\boldsymbol{u}} \left[ \frac{Q_\psi(s, a+\mu\boldsymbol{u})}{\mu} \boldsymbol{u} \right] - \underbrace{\mathbb{E}_{\boldsymbol{u}} \left[ \frac{Q_\psi(s,a)}{\mu} \boldsymbol{u} \right]}_{=0} - \nabla_a Q_\mu^{\pi_\theta}(s,a) + \nabla_a Q_\mu^{\pi_\theta}(s,a) - \nabla_a Q^{\pi_\theta}(s,a) \right\|, \quad (7) \\
&= \left\| \mathbb{E}_{\boldsymbol{u}} \left[ \frac{Q_\psi(s, a+\mu\boldsymbol{u})}{\mu} \boldsymbol{u} \right] - \mathbb{E}_{\boldsymbol{u}} \left[ \frac{Q^{\pi_\theta}(s, a+\mu\boldsymbol{u})}{\mu} \boldsymbol{u} \right] + \nabla_a Q_\mu^{\pi_\theta}(s,a) - \nabla_a Q^{\pi_\theta}(s,a) \right\|, \quad (8) \\
&= \left\| \mathbb{E}_{\boldsymbol{u}} \left[ \frac{Q_\psi(s, a+\mu\boldsymbol{u}) - Q^{\pi_\theta}(s, a+\mu\boldsymbol{u})}{\mu} \boldsymbol{u} \right] + \nabla_a Q_\mu^{\pi_\theta}(s,a) - \nabla_a Q^{\pi_\theta}(s,a) \right\| \\
&\leq \left\| \mathbb{E}_{\boldsymbol{u}} \left[ \frac{Q_\psi(s, a+\mu\boldsymbol{u}) - Q^{\pi_\theta}(s, a+\mu\boldsymbol{u})}{\mu} \boldsymbol{u} \right] \right\| + \left\| \nabla_a Q_\mu^{\pi_\theta}(s,a) - \nabla_a Q^{\pi_\theta}(s,a) \right\| \\
&\leq \frac{1}{\mu} \sqrt{\mathbb{E}_{\boldsymbol{u}}[|Q_\psi(s, a+\mu\boldsymbol{u}) - Q^{\pi_\theta}(s, a+\mu\boldsymbol{u})|^2]} \sqrt{\mathbb{E}_{\boldsymbol{u}}[\|\boldsymbol{u}\|^2]} + \left\| \nabla_a Q_\mu^{\pi_\theta}(s,a) - \nabla_a Q^{\pi_\theta}(s,a) \right\| \\
&= \frac{\sqrt{p}}{\mu} \sqrt{\mathbb{E}_{\boldsymbol{u}}[|Q_\psi(s, a+\mu\boldsymbol{u}) - Q^{\pi_\theta}(s, a+\mu\boldsymbol{u})|^2]} + \left\| \nabla_a Q_\mu^{\pi_\theta}(s,a) - \nabla_a Q^{\pi_\theta}(s,a) \right\|,
\end{aligned}
$$

where we have used the Cauchy–Schwarz inequality in the penultimate line and applied Proposition 1 to reduce equation 7 to equation 8, replacing $\nabla_a Q_\mu^{\pi_\theta}(s,a)$ with $\mathbb{E}_{\boldsymbol{u}} \left[ \frac{Q^{\pi_\theta}(s,a+\mu\boldsymbol{u}) - Q^{\pi_\theta}(s,a)}{\mu} \boldsymbol{u} \right] = \mathbb{E}_{\boldsymbol{u}} \left[ \frac{Q^{\pi_\theta}(s,a+\mu\boldsymbol{u})}{\mu} \boldsymbol{u} \right]$.

Continuing further, taking expectations over $s$, we obtain

$$\frac{(1-\gamma)}{B}\|\hat{\nabla}J(\theta) - \nabla J(\theta)\|$$

$$\leq \mathbb{E}_s\left[\frac{\sqrt{p}}{\mu}\sqrt{\mathbb{E}_{\boldsymbol{u}}[|Q_\psi(s, a + \mu\boldsymbol{u}) - Q^{\pi_\theta}(s, a + \mu\boldsymbol{u})|^2]} + \left.\left\|\nabla_a Q_\mu^{\pi_\theta}(s, a) - \nabla_a Q^{\pi_\theta}(s, a)\right\|\right|_{a=\pi_\theta(s)}\right]$$

$$= \frac{\sqrt{p}}{\mu}\mathbb{E}_s\left[\sqrt{\mathbb{E}_{\boldsymbol{u}}[|Q_\psi(s, \pi_\theta(s) + \mu\boldsymbol{u}) - Q^{\pi_\theta}(s, \pi_\theta(s) + \mu\boldsymbol{u})|^2]}\right]$$

$$+ \mathbb{E}_s\left[\left.\left\|\nabla_a Q_\mu^{\pi_\theta}(s, a) - \nabla_a Q^{\pi_\theta}(s, a)\right\|\right|_{a=\pi_\theta(s)}\right]$$

$$\leq \frac{\sqrt{p}}{\mu}\sqrt{\mathbb{E}_{s,\boldsymbol{u}}[|Q_\psi(s, \pi_\theta(s) + \mu\boldsymbol{u}) - Q^{\pi_\theta}(s, \pi_\theta(s) + \mu\boldsymbol{u})|^2]}$$

$$+ \mathbb{E}_s\left[\left.\left\|\nabla_a Q_\mu^{\pi_\theta}(s, a) - \nabla_a Q^{\pi_\theta}(s, a)\right\|\right|_{a=\pi_\theta(s)}\right]$$

$$= \frac{\sqrt{p}}{\mu}\varepsilon_{\mu,\psi}^{\pi_\theta} + \mathbb{E}_s\left[\left.\left\|\nabla_a Q_\mu^{\pi_\theta}(s, a) - \nabla_a Q^{\pi_\theta}(s, a)\right\|\right|_{a=\pi_\theta(s)}\right],$$

where we have used Jensen in the third line (the root function is concave). The rest of the theorem follows trivially from Proposition 1, which is based on the results invoked from Nesterov and Spokoiny (2017), and we are done.

### A.4 A Relaxed Notion of Compatibility via $\epsilon$-Compatible Critics

We may formalize the notion of approximate ($\epsilon$-)compatibility put forth in this work by introducing the following definition:

**Definition 1** ($\epsilon$-**Compatibility**). *Let $\epsilon \geq 0$ and $\theta \in \Theta$. We say that a critic $\hat{Q}$ is (pointwise) $\epsilon$-compatible at $\theta$ if the corresponding deterministic policy gradient approximation $\hat{\nabla}$ satisfies*

$$\|\hat{\nabla}J(\theta) - \nabla J(\theta)\| \leq \epsilon.$$

*In this case, we also say that $\hat{\nabla}J$ is $\epsilon$-compatible at $\theta$.*

We note that the error $\epsilon$ may depend on the policy parameters $\theta$, as well as on additional parameters used in the construction of $\hat{Q}$ and $\hat{\nabla}J$ (e.g., $\mu$ and $\psi$ in Theorem 1).

This formulation generalizes the classical notion of compatibility from Silver et al. (2014, Section 4.3). In particular, if a critic is compatible in the sense of Silver et al. (2014), then it is trivially $\epsilon$-compatible with $\epsilon = 0$ at every $\theta \in \Theta$. However, the converse does not hold unless $\epsilon = 0$: a critic that is $\epsilon$-compatible for $\epsilon > 0$ need not satisfy exact compatibility.

Compared to the conventional approach of using the gradient of a fitted $Q$-network trained on finite samples, the advantage of $\epsilon$-compatibility lies in its theoretical consistency across a broader class of approximators. This is confirmed by Theorem 1, and has direct algorithmic relevance. In this sense, designing a critic that satisfies $\epsilon$-compatibility is theoretically preferable—and practically beneficial, as our empirical results indicate—due to the following reasons:

- The classical compatibility condition in Silver et al. (2014) only applies to critics whose gradients are linear in their parameters, which severely restricts model expressivity.

- More generally, using gradients of arbitrary fitted critics does not guarantee compatibility unless additional constraints (e.g., smoothness or regularization) are imposed. As illustrated in Appendix A.2, our zeroth-order, Gaussian smoothing-based gradient estimator implicitly imposes such constraints, and is provably $\epsilon$-compatible by Theorem 1.

### A.5 Compatible Policy Gradient with Learnable Exploration Noise

Recent studies have explored learning exploration noise through neural networks. For instance, the closest off-policy RL framework is used by Saglam and Kozat (2023), where the noise term, typically sampled from

an isotropic Gaussian, is replaced with the output of a neural network initialized as the policy network. This *exploration policy* outputs the noise based on the observed state, aiming to maximize uncertainty (i.e., where the loss of the $Q$-network is highest). A natural question that arises is what would happen if we combined such methods with oCPG.

We do not believe it would negatively impact performance because, from the policy perspective, exploratory action noise remains just noise, regardless of the distribution from which it is sampled. Unless the norm of the noise increases significantly, the computed PG would not differ greatly from that computed with Gaussian noise. However, we did not include any learnable exploration method in our experiments, as our focus is solely on studying the pure effects of PG computation, and any additional technique could have confounded our conclusions.

## B  Experimental Details

In Section B.1, we provide details of the hyperparameters and network architectures. Additionally, the experimental setup, such as simulation and performance assessment, is examined in Section B.2.

### B.1  Hyperparameters and Neural Networks

**Architecture**  All algorithms use two $Q$-networks following the Clipped Double $Q$-learning algorithm, and one actor network. Each network consists of two hidden layers with 256 units each, using ReLU activations. The $Q$-networks process state-action pairs $(s, a)$ and output a scalar value. The actor network inputs state $s$ and outputs an action $a$ using a `tanh` activation, scaled to the action range of the environment.

Table 2: Hyperparameters used in the experiments

| Hyperparameter | Value |
|---|---|
| Optimizer | Adam |
| Learning rate (all networks) | $3 \times 10^{-4}$ |
| Learning rate (CPG actor only) | $5 \times 10^{-5}$ |
| Minibatch size | 256 |
| Discount factor $\gamma$ | 0.99 |
| Target update rate | 0.005 |
| Initial exploration steps | 25,000 |
| Exploration noise $\mu$ | 0.1 |
| TD3 target policy noise $\sigma$ | 0.2 |
| TD3 policy noise clipping | $(-0.5, 0.5)$ |
| SAC log-standard deviation clipping | $(-20, 2)$ |
| SAC log constant | $10^{-6}$ |
| SAC reward scale (except Humanoid) | 5 |
| SAC reward scale (Humanoid) | 20 |

**Hyperparameters**  The Adam optimizer (Kingma and Ba, 2015) is used for training the networks, with a learning rate set to $3 \times 10^{-4}$. When sampling from the replay buffer, minibatches of 256 samples are employed. We update the target networks using polyak averaging, with a rate of $\tau = 0.005$. This process follows the update rule: $\theta' \leftarrow 0.995 \times \theta' + 0.005 \times \theta$.

**Hyperparameter Optimization**  For SAC, hyperparameter optimization was performed, particularly for the Swimmer task, as the original paper did not specify the reward scale. We tested reward scales of $\{5, 10, 20\}$ and found that a factor of 5 delivered the best performance in this environment. Beyond this, all algorithms strictly followed the parameter settings from their papers or the latest GitHub versions of their code. Specifically, SAC followed the hyperparameters from its original paper, with the exception of

increasing exploration steps (i.e., a purely exploratory policy with no learning) to 25,000. For TD3, we used the author's GitHub repository[2], which introduced slight parameter changes compared to the paper. Notably, this version increased the start steps to 25,000 and the batch size to 256 across all environments, resulting in improved performance. The full hyperparameter settings are provided in Table 2.

### B.2 Experimental Setup

**Simulation Environments** All agents are evaluated using continuous control benchmarks from the MuJoCo[3] physics engine, accessed through OpenAI Gym[4] with `v3` environments. We maintained the original state-action spaces and reward functions in these environments to ensure reproducibility and fair comparison with existing empirical results. Each environment features a multi-dimensional action space within the range of [-1, 1], except for Humanoid, which has a range of [-0.4, 0.4].

**Terminal Transitions** In computing the target $Q$-value, we apply a discount factor of $\gamma = 0.99$ for non-terminal transitions, and zero for terminal ones. A transition is considered as terminal only if it ends due to a termination condition, such as a failure or exceeding the time limit or fall of the agent.

**Evaluation** Evaluations are performed every 1000 time steps, with each evaluation representing the average reward across 10 episodes. These evaluations use the deterministic policy from oCPG and TD3 without exploration noise, and the deterministic mean action from SAC. To minimize variation due to different seeds, a new environment with a fixed seed (the training seed plus a constant) is used for each evaluation. This approach ensures consistent initial start states for each evaluation.

**Visualization of the Learning Curves** Learning curves, illustrating performance, are plotted as averages from 10 trials. A shaded region around each curve represents the 95% confidence interval across these trials. For enhanced visual clarity, the curves are smoothed using a sliding window averaging over a number evaluations determined by 0.05 of the length of the curves.

### B.3 Computational Resources

Our experiments were conducted on a high-performance computing system with an AMD Ryzen Threadripper PRO 3995WX processor (64 cores) and 512 GB of RAM. Neural network training utilized a single NVIDIA RTX A6000 GPU with 48 GB of VRAM, providing sufficient computational resources for our tasks.

## C Complementary Results

### C.1 Numerical Comparison Under Different Statistical Tests

In the main body, we consider four different statistical tests. To ensure a comprehensive statistical comparison, we provide brief descriptions of each method and their approach to conducting statistical analysis.

- **Welch's *t*-test:** It accounts for potential differences in variances between the two groups being compared, providing a more robust analysis when variances are unequal. Although our sample sizes are identical, differences in performance variance between our method and the baselines could exist, making Welch's *t*-test the most appropriate choice to ensure the validity of our statistical significance claims.

- **Student's *t*-test:** Unlike Welch's test, it compares the means of two independent groups under the assumption that the variances are equal. It is suitable when the distribution of the data is normal, and the variances between groups do not differ significantly.

---

[2]`https://github.com/sfujim/TD3`
[3]`https://mujoco.org/`
[4]`https://www.gymlibrary.ml/`

- **Paired $t$-test:** It is used to compare the means of two related groups, such as measurements taken from the same subjects under different conditions. It is ideal for designs where each subject serves as their own control, thereby reducing the impact of variability. Specifically, we aim to reflect the sole difference between oCPG and TD3, which is the PG computation that will serve as the subject's *own control.*

- **Wilcoxon rank-sum test:** To complement the assumption of Gaussian-distributed sample data, we employ this non-parametric alternative to the independent $t$-test. It compares two independent groups without assuming a normal distribution. It is particularly useful when the data is skewed or contains outliers.

We observe that the statistical significance of the improvement offered by CPG is consistent across different tests, as highlighted in Table 1. This strengthens the validity of our empirical studies and underscores the benefits of our proposed method.

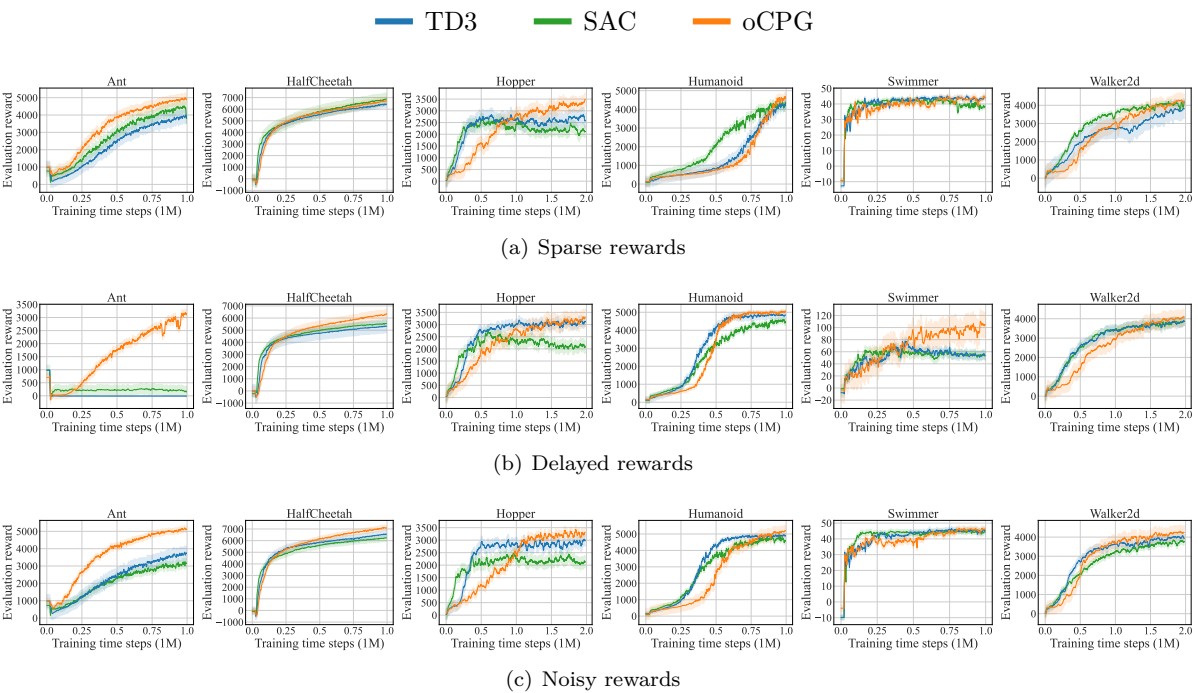

Figure 3: Learning curves for the benchmark MuJoCo environments, averaged over 10 random seeds, under imperfect environment conditions: (a) the agent observes the true reward with a probability of 0.5, otherwise zero; (b) rewards are delayed by 10 time steps; and (c) rewards are perturbed by a zero-mean Gaussian noise with a standard deviation equal to 0.1 of the reward range in the replay buffer. The shaded area represents the 95% confidence interval of the mean performance.

## C.2 Learning Curves for Imperfect Environmental Conditions

**Uncertainty in the Environments**  To evaluate the robustness of oCPG, we simulate imperfect environmental conditions by introducing uncertainty. This includes considering scenarios with noisy, sparse, and delayed rewards. In the noisy reward scenario, each reward is subject to perturbation with random noise. For the sparse reward setup, the agent observes the reward with a probability less than 1; otherwise, it receives a reward of zero. In the delayed setting, the agent observes the reward after a fixed number of time steps.

**Setup**  The delayed reward setting delays the instantaneous rewards received by the agent by 10 time steps. For noisy rewards, we monitor the range of rewards collected by the agent, specifically the minimum and maximum values. The reward-perturbation noise is then sampled from a normal distribution $\mathcal{N}(0, 0.1 \cdot (r_{\max} - r_{\min}))$, where $r_{\max}$ and $r_{\min}$ are the maximum and minimum of the rewards in the replay buffer. In the case of sparse rewards, with a random probability of 0.5, the agent either observes the actual instantaneous reward or zero.

