# OpenReview forum: "Compatible Gradient Approximations for Actor-Critic Algorithms"
_TMLR — Rejected by TMLR_

### Review · Reviewer_pnxs · 2025-09-24

**Summary Of Contributions:**

This paper proposes a continuous control actor-critic algorithm called off-policy Compatible Policy Gradient (oCPG in short) using a zeroth order approximation of the action-value gradient  in the action space (via a two-point stochastic gradient estimate).

The main idea is to bypass the direct computation of the action-value gradient by resorting to zeroth order approximation while still addressing the compatibility requirement in Deterministic Policy Gradient (DPG).
In particular, the distance between the true deterministic policy gradient and the proposed approximation is controlled.
The performance of the proposed algorithm is evaluated on the standard OpenAI Gym continuous control tasks.

**Strengths**

- Policy gradient methods with function approximation are popular methods which are central to the success of RL in practice. The analysis of these methods is still an active research topic.

- The exposition is globally clear and easy to follow.


**Weaknesses**

- **Main concern:**
The idea developed in this paper has already been proposed and investigated in details in prior work (see Vemula et al. 2019, Kumar et al. 2020, references below). These papers are not discussed nor cited in the literature review. In particular, the DPG zeroth order approximation (eq. 5 in the submission) which essentially captures the main idea of the paper, has been previously proposed as is in Lemma 1 in the second paper below as well as in the first paper below (Algorithm 3 therein). Moreover, both papers below also provide a convergence analysis (see e.g. section 4 in Kumar et al.) whereas the present submission lists it as future work. Up to the additional perturbation representation error (which is a straightforward extension compared to existing work), Theorem 1 is also covered by Theorem 1 in Kumar et al. (including its proof).

--*Vemula et al. Contrasting Exploration in Parameter and Action Space: A Zeroth-Order Optimization Perspective. AISTATS 2019.*

--*Kumar et al. 2020. Zeroth-order Deterministic Policy Gradient. https://arxiv.org/pdf/2006.07314*

Published version: Kumar et al. Actor-only Deterministic Policy Gradient via Zeroth-order Gradient Oracles in Action Space. 2021 IEEE International Symposium on Information Theory (ISIT).

- **Dimension dependence and comparison to first-order PG:** PG methods are especially popular for high-dimensional settings. Limitations of the zeroth order approach in terms of dimension of the action space are not discussed. As well-known in optimization, the performance of zeroth order methods deteriorates significantly with the dimension of the underlying decision variable space in contrast to first-order (gradient-based) methods. I think the paper should explicitly discuss this limitation.  See e.g. scaling in p (dimension) in Proposition 1. The paper mentions for instance in p. 4-5 that the action space is 'usually of substantially lower dimension as compared with the dimension of the corresponding parameter space' but does not mention the dimension dependence limitation. Note that the work of Kumar et al. mentioned above does so.

- **Compatible function approximation error:**  Such a quantity has also been introduced and used in several theoretical RL works in the literature (see e.g. references below). How do the definition and result of Theorem 1 compare to existing works? The paper considers a zeroth order estimate using smoothing which differs from these existing works and neural networks are not necessarily linear in the parameter $\psi$. I think it would be interesting to discuss how they do compare in the linear case for instance.

-- *Agarwal et al. On the Theory of Policy Gradient Methods: Optimality, Approximation, and Distribution Shift. JMLR 2021.*

-- *Alfano et al. A Novel Framework for Policy Mirror Descent with General Parameterization and Linear Convergence. Neurips 2023.*

- **Sample complexity:** It is unclear how the sample complexity of the proposed zeroth order method compares to existing benchmark PG algorithms in the paper (see Q2 below).

- **Continuous state-action space (presentation, minor):** In the continuous state action space which is the focus of the paper, MDPs are not rigorously introduced in the paper. In such a setting, the probability to transition from a given state to another state is actually always zero (because the probability for a real-valued random variable to be equal to any given real value is in general zero, unlike for discrete valued random variables). Same comment for the Markov property which cannot be defined properly using only single states (as in the technical preliminaries of section 3).
One needs to introduce borel sets (measurable sets) over the state and action spaces and define a Markov transition kernel. See e.g. section 2.3.2, Puterman '05 (Markov Decision Processes: Discrete Stochastic Dynamic Programming) p.28-29. Otherwise, the definitions are not meaningful for the continuous state action space setting.

**Additional Comments:**

**Questions**

1. Can you further clarify using your notations how do existing methods (which are first-order based rather than zeroth order-based) approximate gradients when compatibility conditions are not met? The paper seems to argue that existing methods are not principled, but is it possible to propose a principled first-order method rather than a zeroth-order one? Is there a particular oracle requirement or a stronger feedback required compared to the proposed zeroth order approach if that is possible?
The paper discusses some of these aspects in section 4.1, mentioning that compatibility conditions will be violated when using neural networks. Still, can we also define a suitable compatibility function approximation error in that case similarly to the paper's proposal in Theorem 1.

2. Can you compare the proposed method with TD3, SAC, PPO in terms of sample complexity for fair comparison? Experiments focus on reward performance and show that oCPG is competitive with other existing PG algorithms and shows some improved performance in some settings. However, I expect sample complexity to be more important, due for instance to the dimension dependence induced by random action selection. I think it would be useful to discuss this possible limitation of the zeroth order approach (see my comment above regarding dimension dependence).

3. Is there a way to show the (approximate) compatibility of the proposed policy gradient estimate compared to (a) the true policy gradient and (b) the policy gradients used in the literature (under their simplest heuristic form) on very simple toy examples (even bandit settings if that is still meaningful)?  I think that would really be more convincing to support the main motivation of the approach and the question of approximate compatibility under function approximation.

4. Are there settings or RL applications where the zeroth order approach is the only possible approach for policy optimization, i.e. action-value gradients cannot be computed using first-order information? (E.g. in black box optimization problems, this is the case). I believe that would further strengthen the motivation of the work. That being said, since oCPG does still use some tricks (clipped double Q learning for critic estimation for instance), the algorithm is not fully zeroth order.

 **Minor/typos:**

- p. 5: 'that near-solves equation 1'. Equation 1 is not a problem here, I guess you mean optimize the objective in equation 1.
- p. 6: typo: intruduced


**Additional comment**

Some existing works (see reference below) argue in the literature that approximation error (under function approximation) might not be a key quantity for characterizing global convergence of PG methods. Some comments regarding this and the focus of the paper on the proposed PRE error would be useful.

-- *Mei et al. Ordering-based Conditions for Global Convergence of Policy Gradient Methods. Neurips 2023.*

-- *Lin et al. Rethinking the Global Convergence of Softmax Policy Gradient with Linear Function Approximation. 2025, arxiv https://arxiv.org/pdf/2505.03155.*

**Audience:**

Yes

**Audience Explanation:**

Policy gradient methods with function approximation are popular methods which are central to the success of RL in practice. The analysis of these methods is still an active research topic.

**Claims And Evidence:**

No

**Claims Explanation:**

See my main concern above regarding the claim of novelty compared to prior work using zeroth order approximation in action space for policy optimization.

**Requested Changes:**

The following points are crucial:
- Discuss the two crucial related works in details (see main concern) to highlight the contributions of the present work (at this stage, only experiments seem to be new).
- Comment on sample complexity of the proposed zeroth order method compared to existing first-order PG methods.
- Discuss limitations of zeroth order optimization in terms of dimension dependence. How does the performance depend on the dimension of the action space in the experiments?

See additional comments and questions below to be addressed.

---

> ### Author Response · Authors · 2025-10-20
> **Response — Part I**
>
> Thank you for your constructive comments and insightful suggestions.
>
> ---
>
> ### **Forthcoming:** Convergence Analysis
>
> Before we begin our responses, we want to note that we are currently finalizing a detailed convergence analysis of oCPG and will post the complete proof during the week of October 20.
>
> This analysis explicitly incorporates the effects of the replay buffer (capturing off-policy discrepancies) and the Q-network approximation via the Clipped Double Q-learning algorithm—an integration that, to the best of our knowledge, has not been formally analyzed in the literature.
>
> We hope the reviewer is fine with this timeline.
>
> ---
>
> ### **Main Concern:** Clarifying Novelty and Relation to Prior Work
>
> Thank you for raising this important point. The two-point smoothing identity is indeed standard in zeroth-order optimization and also used in ZDPG (Kumar et al., 2020). Our contribution, however, is not the identity itself but its integration into a practical, deep actor–critic framework. Specifically, we introduce a new error notion—the perturbation representation error (PRE)—and an $\varepsilon$-compatibility analysis that explicitly accounts for the bias induced by critic approximation.
>
> ZDPG is an *actor-only* method that assumes oracle access to
> 1. unbiased evaluations of $Q^{\pi_\theta}(s,a)$
> 2. samples from the discounted state distribution $(1-\gamma)\rho^{\pi_\theta}$, obtained through random-horizon rollouts.
>
> Their update rule is
>
> $
> g\_t = \frac{1}{1-\gamma} \nabla\_\theta \pi\_\theta(s\_t) \frac{\widehat Q(s\_t,\pi\_\theta(s\_t)+\mu u\_t)-\widehat Q(s\_t, \pi\_\theta(s\_t))}{\mu} u\_t,
> $
>
> where $\widehat Q$ is computed by **simulating the environment twice** (at $a$ and $a+\mu u$ to a geometrically distributed horizon (Alg. 2 in their paper), and $s_t$ is drawn from a separate geometric-horizon sampler (Alg. 3). Their Theorem 1 bounds $ \|\nabla J(\theta)-\widehat{\nabla}J(\theta)\|$ solely in terms of smoothing error under Lipschitz assumptions, and Theorem 2 provides rates for biased SGD on the actor. Importantly, *no learned critic is involved*.
>
> In contrast, our work develops a full deep actor–critic framework. We replace oracle $Q$ evaluations with a learned critic $Q_\psi$ and analyze the induced bias through the PRE, which quantifies the discrepancy between $Q_\psi$ and $Q^{\pi_\theta}$ at perturbed actions. This term is absent in ZDPG, as their approach never approximates $Q^{\pi_\theta}$. Consequently, our Theorem 1 extends their analysis to the function-approximation regime and provides a concrete prescription for training $Q_\psi$ to control actor-gradient bias by minimizing PRE.
>
> In summary, Eq. (5) in our paper is not a restatement of Lemma 1 from ZDPG but its *actor–critic extension*, where oracle $Q^{\pi_\theta}$ is replaced by a learned critic $Q_\psi$. Theorem 1 then quantifies the resulting bias through PRE. This extension—together with our off-policy deep RL formulation—constitutes the conceptual and technical novelty of our work relative to ZDPG and to prior zeroth-order analyses such as Vemula et al. (2019), which focus on parameter- versus action-space exploration in *finite*-horizon settings.
>
> In the revised paper, we have cited both works and incorporated these distinctions into the extended Related Work section.
>
> ---
> ### **Sample Complexity:** Action Dimension Dependence and Comparison to First-Order PG
>
> Thank you for raising these closely connected points. The dependence on the action dimension $p$ is indeed a limitation of zeroth-order methods. Because our gradient estimator relies on random perturbations in the $p$-dimensional action space, its variance scales as $\mathcal{O}(p/N)$, where $N$ is the batch size. Thus, obtaining an $\varepsilon$-accurate estimate requires $N = \mathcal{O}(p/\varepsilon^2)$ samples, reflecting linear dependence on $p$.
>
> This trade-off contrasts with first-order policy gradient methods, which do not introduce such dimension dependence but can yield biased gradients when compatibility conditions are not met. In effect, our approach exchanges dimension-free sample complexity for unbiased and provably compatible gradients.
>
> In practice, the environments we study have relatively small action dimensions (ranges from 2 to 17 in the MuJoCo environments we consider), where this variance penalty remains mild. We acknowledge, however, that this limitation becomes more pronounced as $p$ grows, consistent with known results on zeroth-order optimization [1, 2].
>
> To improve clarity, we have added a technical discussion of this limitation at the end of Section 4.2.1 ("Further Remarks"). We have also added the action space dimensionality of each environment to Table 1 and expanded the experiments discussion to note that the practical impact of $p$ is modest in the examined settings.

---

> ### Author Response · Authors · 2025-10-20
> **Response — Part II**
>
> ### **Compatible Function Approximation Error**
>
> Thank you for pointing out this connection. We clarify that Theorem 1 addresses a different but related source of error. In Agarwal et al. (2021) [3] and Alfano et al. (2023) [4], the error arises when the critic violates compatibility conditions in first-order methods (e.g., absence of linear gradients, as summarized in Appendix A.1), with bounds derived under linear or general parameterizations.
>
> In contrast, our result characterizes the error of a zeroth-order surrogate that replaces $\nabla_a Q$ with stochastic two-point perturbations in the action space. The approximation error is quantified by the perturbation representation error (PRE), which is distinct from but analogous to the classical compatibility error in the bound. In the linear critic setting, minimizing PRE yields an error bound consistent with the standard compatible function approximation error, showing that our result subsumes the linear case. Beyond this setting, Theorem 1 extends the analysis to nonlinear critics and zeroth-order estimation, where classical compatibility assumptions no longer apply.
>
> To better highlight this distinction, we have added a paragraph to Related Work (as the second paragraph).
>
> ---
>
> ### **Continuous State-Action Space** (presentation, minor)
>
> Thanks for bringing this. We have now improved the corresponding part of Technical Preliminaries for technical rigor.
>
> ---
>
> ### **Question 1**
>
> _**How do existing first-order (FO) methods behave without compatibility?**_ DPG updates the actor with
>
> $
> g\_{\text{FO}}(\theta) = \frac{1}{1-\gamma}\mathbb{E}\_{s\sim\rho\_{\pi\_\theta}}
> \left[\nabla\_\theta \pi\_\theta(s)\,
> \nabla\_a Q\_\omega(s,a)\big|\_{a=\pi\_\theta(s)}\right],
> $
>
> which is unbiased only under the *compatibility* conditions recalled in Appendix A.1:
>
> $
> \nabla\_a Q\_\omega(s,a)\big|\_{a=\pi\_\theta(s)} = \nabla\_\theta \pi\_\theta(s)^\top \omega,
> $
>
> together with $\omega$ minimizing the gradient mean squared error
>
> $
> \mathbb{E}\_s \left[ \big\| \nabla\_a Q\_\omega(s,a)\big|\_{a=\pi\_\theta(s)} - \nabla\_a Q^{\pi\_\theta}(s,a)\big|\_{a=\pi\_\theta(s)} \big\|^2 \right].
> $
>
> When the critic is approximated by a deep nonlinear network, $g\_{\text{FO}}(\theta)$ can be biased and may cease to be an ascent direction (see Appendix A.1 for the conditions; see Appendix A.2 for the practical failure mode).
>
> _**Can we design a principled FO alternative?**_ One could posit the FO "compatibility error":
>
> $
> \varepsilon^{\text{FO}}\_{\pi\_\theta,\omega} \coloneqq \sqrt{\mathbb{E}\_s\big[\| \nabla\_a Q\_\omega(s,a) - \nabla\_a Q^{\pi\_\theta}(s, a)\|^2\big]},
> $
>
> and try to train $\omega$ to minimize it. However, this requires access to the oracle quantity $\nabla\_a Q^{\pi\_\theta}$ that standard RL feedback does not reveal (e.g., we cannot get a feedback on the gradient from, say, rewards in the replay buffer); equivalently, the classical compatibility condition *already* asks to minimize the gradient MSE to the *true* action-gradient (App. A.1, condition 2), which is infeasible in deep settings without stronger feedback. The other classical route—architecturally enforcing $\nabla\_a Q\_\omega=\nabla\_\theta\pi\_\theta^\top\omega$ (App. A.1, condition 1)—is violated when deep networks are used.
>
> _**What our PRE measures and why zeroth-order (ZO) optimization helps.**_ We define the PRE as a value-space mismatch at perturbed actions:
>
> $
> \varepsilon^\text{PRE}\_{\pi\_\theta, \mu,\psi} \coloneqq \sqrt{\mathbb{E}\_{s, u}\big[\big\|Q\_{\psi}(s,\pi\_{\theta}(s)+\mu u)-Q^{\pi\_{\theta}}(s,\pi\_{\theta}(s)+\mu u)\big\|^{2}\big]},
> $
>
> Crucially, PRE *does not* involve $\nabla\_a Q^{\pi\_\theta}$. Theorem 1 then gives
>
> $
> \big\|\widehat{\nabla}\_{\mu,\psi}J(\theta)-\nabla J(\theta)\big\| \le \frac{B}{1-\gamma} \left[ \frac{\varepsilon^{\text{PRE}}\_{\pi\_\theta,\mu,\psi}}{\mu}\sqrt{p} + \mathbb{E}\_s\big[\|\nabla\_a Q^{\pi\_\theta}\_\mu-\nabla\_a Q^{\pi\_\theta}\|\_{a=\pi\_\theta(s)}\big] \right],
> $
>
> and if $Q^{\pi\_\theta}$ is $G$-smooth, the second term is $\le G\mu$, yielding $\frac{B\sqrt{p}}{1-\gamma}\big(\epsilon^{\text{PRE}}\_{\pi\_\theta,\mu,\psi}/\mu + G\mu\big)$.
>
> Thus, with small $\mu$ and a critic trained to reduce the *value-space* mismatch at perturbed actions, the gradient error is controlled without requiring access to the true action-gradient.
>
> _**Conclusion.**_ A "principled FO fix" would either *(i)* enforce the linear gradient form of App. A.1 (restrictive), or *(ii)* require an oracle for $\nabla\_a Q^{\pi\_\theta}$ to minimize a gradient-space error—requires stronger feedback than what standard transitions provide (e.g., reward). Our ZO surrogate sidesteps both: it operates on critic *values* at perturbed actions (PRE), and Theorem 1 makes the resulting bias fully transparent in terms of $(\mu,\varepsilon^{\text{PRE}})$ and a standard smoothing gap.

---

> ### Author Response · Authors · 2025-10-20
> **Response — Part III**
>
> ### **Question 2**
> Please see our response above (Part I). We have added a paragraph to Section 4.2.1. ("Further Remarks") to discuss this limitation under sample complexity.
>
> ---
>
> ### **Question 3**
> We appreciate this helpful suggestion. To address this, we implemented a controlled study using a linear quadratic regulator (LQR), where exact Q-values and gradients can be computed in closed form.
>
> _**Experimental Setup.**_ The environment is a discrete-time LQR with dynamics
>
> $
> s_{t+1} = A s\_t + B a\_t + w\_t,\quad w\_t \sim \mathcal{N}(0,0.1^2I),
> $
>
> with the cost function
>
> $
> c(s,a) = s^\top Q\_c s + a^\top R\_c a,\quad Q\_c = I,\ R\_c = \mathrm{diag}(0.01,0.1,1,0.01,0.1,1),
> $
>
> and optimal solution obtained from the discrete algebraic Riccati equation. Therefore, we have the true Q-value for state-action pairs in hand.
>
> We initialize the policy to be linear, $\pi\_\theta(s) = K\_\theta s$. To construct the training data, we generate a static dataset of 500 state--action pairs using a behavior policy $a = 0.7K^* s$, where $K^*$ is the optimal linear feedback gain obtained from the Riccati solution. This choice ensures the dataset covers near-optimal actions while still allowing sufficient exploration. Using a static dataset highlights the finite-sample regime and guarantees that both methods are evaluated under identical conditions without the confound of on-policy data collection.
>
> For each sampled pair, we also include perturbed neighbors $(a \pm \mu v)$, with $\mu = 0.005$ and $u$ drawn uniformly from the unit sphere, so that local value information around each action is available. The critic is a two-layer ReLU network (256 hidden units per layer) trained by supervised regression on the true Q-values. This architecture follows standard practice for critic initialization in actor--critic algorithms such as TD3 and SAC.
>
> _**Gradient Estimators.**_ We consider:
>
> - the true gradient $\nabla J(\theta) = \frac{1}{N}\sum\_{i=1}^N\nabla \pi(s\_i)\nabla\_a Q^*(s\_i, a\_i)\vert\_{a\_i = \pi\_\theta(s\_i)}$,
> - the DPG estimate $\nabla \hat{J}\_\text{DPG}(\theta) = \frac{1}{N}\sum\_{i=1}^N \nabla \pi(s\_i) \nabla Q\_\psi(s\_i, a\_i)\vert\_{a\_i = \pi\_\theta(s\_i)}$,
> - and the symmetric zeroth-order estimator
>
> $
> \hat{J}\_{\text{ZOO}}(\theta)=\frac{1}{N}\sum\_{i=1}^N \nabla \pi(s\_i)\,\left.\left[\frac{Q\_\psi(s\_i, a\_i+\mu u\_i) - Q\_\psi(s\_i, a\_i - \mu u\_i)}{2\mu}\,u\_i\right]\right|\_{a\_i=\pi\_\theta(s\_i)},
> $
>
> where $N$ is the batch size, and compare:
>
> - Relative gradient estimation error of DPG: $\frac{\|\nabla J(\theta) - \nabla \hat{J}\_\text{DPG}(\theta)\|}{\|\nabla J(\theta)\|}$
> - Relative gradient estimation error of CPG: $\frac{\|\nabla J(\theta) - \nabla \hat{J}\_\text{ZOO}(\theta)\|}{\|\nabla J(\theta)\|}$
>
> _**Results.**_
> We provide the corresponding Q-value and gradient error curves in the revised paper and in our *anonymous* [GitHub repository](https://anonymous.4open.science/r/compat-grad-approx/figs/lqr_example/gradient_errors.png). The key observations are summarized below.
>
> - The critic achieves very low Q-value estimation error across training, confirming that value approximation itself is not the limiting factor.
> - Relative gradient error -- DPG starts near 1.0 and gradually rises to about 2.4, indicating increasing drift of critic derivatives, while the ZO estimator remains stable near 1.0 throughout training.
>
> _**Discussion.**_
> These results confirm that even when value approximation is accurate, the analytic gradients of a ReLU critic can drift significantly due to finite-sample and nonsmooth effects, leading to inflated norms and unstable DPG estimates.
>
> In contrast, our method only requires accurate evaluations of the smoothed actions to yield reliable gradients. In this experiment, this is achieved by including perturbed (smoothed) actions in the static dataset and fitting the critic to them. In practice (e.g., main experiments in the paper), the same effect arises naturally because Gaussian exploration noise generates smoothed actions (we use the same $\mu$); the replay buffer therefore already contains such perturbed samples, ensuring that the critic is trained on the right distribution for our estimator.
>
> Lastly, in an off-policy setting (as in the main experiments), the distributional shift inherent to off-policy learning would introduce an additional error term, visible as an offset in the gradient error curve. This effect is unavoidable unless correction techniques such as importance sampling are applied. As this issue is orthogonal to our study, we leave it for future work; a discussion already appears in the main paper.
>
> Thank you again for suggesting this addition. We now report this experiment in the newly added Section 5.2, "Linear Quadratic Regulator Example." We believe it strengthens the overall motivation of the paper and further clarifies the advantages of our approach.

---

> ### Author Response · Authors · 2025-10-20
> **Response — Part IV**
>
> ### **Question 4**
>
> We use "zeroth-order" to refer *specifically* to the estimation of the action-value gradient $\nabla\_a Q$. The algorithm still trains the critic with standard TD updates (gradients w.r.t. $\psi$). Clipped Double Q-learning is a *training objective for the critic* that reduces overestimation bias; it does *not* supply first-order information about $\nabla\_a Q$. Hence, its use does not negate that the *policy-gradient estimate* is zeroth-order in the action space, while critic fitting remains standard.
>
> There do exist RL settings where first-order actor updates are ill-defined, making a zeroth-order surrogate the only viable option. Examples include:
>
> - **Non-differentiable action post-processing:** when executed actions are projected or clipped by an additional layer post-processing layer/mechanism, $Q^\pi(s,a)$ is typically non-smooth in $a$, so $\nabla\_a Q^\pi$ does not exist.
> - **Discontinuous rewards or dynamics:** threshold penalties or contact events lead to piecewise or discontinuous $Q^\pi$, where first-order gradients are undefined or uninformative.
> - **Non-differentiable critics:** critics based on trees, quantization, or other non-smooth architectures cannot provide $\nabla\_a Q\_\omega$ even though they can be evaluated at given actions.
>
> In all such cases, standard first-order updates break down, while our surrogate—requiring only critic evaluations at perturbed actions—remains applicable and provides a principled update direction.
>
> ---
>
> ### **Minor**
>
> Thank you for noticing. We have now fixed them.
>
> ---
>
> ### **Additional Comment**
>
> Thank you for suggesting these references. Both of these studies analyze global convergence of softmax PG/NPG under linear function approximation in finite-arm bandit settings. Their main conclusion is that approximation error is not the decisive factor for global convergence in those settings, with ordering conditions playing the central role instead.
>
> Our focus is different: PRE quantifies the bias in deterministic, continuous-action actor–critic updates, where policies and critics are parameterized by neural networks and convergence analyses of the type studied in the cited works do not directly apply. We therefore view these results as complementary rather than contradictory, since they address global convergence in discrete softmax policies, whereas our PRE addresses local gradient fidelity under continuous deterministic policies.
>
> ---
>
> ### **References**
>
> [1] Yurii Nesterov and Vladimir Spokoiny. _Random gradient-free minimization of convex functions._ Foundations of Computational Mathematics.
>
> [2] John C. Duchi, Michael I. Jordan, Martin J. Wainwright, and Andre Wibisono. Optimal rates for zero-order convex optimization: _The power of two function evaluations._ IEEE Transactions on Information Theory.
>
> [3] Alekh Agarwal, Sham M. Kakade, Jason D. Lee, and Gaurav Mahajan. _On the theory of policy gradient methods: optimality, approximation, and distribution shift._ JMLR.
>
> [4] Carlo Alfano, Rui Yuan, and Patrick Rebeschini. _A novel framework for policy mirror descent with general parameterization and linear convergence._ NeurIPS, 2023.

---

> > ### Comment · Reviewer_pnxs · 2025-11-02
> > **Response to authors' rebuttal**
> >
> > I thank the authors for their detailed responses. While the rebuttal addresses some of my concerns and I appreciate the efforts of the authors in providing additional experiments and a new theoretical result in particular, I still have many concerns and I think this paper deserves significant rewriting to discuss prior work adequately, clarify the positioning of the paper,  report complete new results with their assumptions, relevant discussions and discuss new experiments.
> >
> > The revised manuscript does not seem to be accessible under its new form (e.g. I cannot find the additional experiment on LQR). I am not sure if this is a restriction of openreview.
> >
> > Below are more detailed comments.
> >
> > **New convergence result:**
> >
> > - The first-order stationary guarantee for smooth non-convex optimization is standard in optimization. Moreover, it has been previously proved (without critic) in the two papers I mentioned in my original review (see Lemma 3, Theorem 2 in Kumar et al. 2022, see also Theorem 4.1 in Vemula et al.) This is not discussed in the paper nor in the rebuttal. The analysis provided in the rebuttal simply extends existing analysis by adding 4 additional error terms which are not controlled but remain as additive terms in the final first-order-stationary guarantee. These terms depend on $T$, so they do not provide any form of guarantee for the convergence rate as it cannot be concluded if the gradient norms do converge to zero. Overall, this additional result does not provide additional useful insights in my opinion, if the errors cannot be clearly controlled under reasonable assumptions. Some of them are function approximation error floors due to the power of the function approximators, still this can be made more explicit even for these terms.
> >
> > - It is assumed that the objective is both Lipschitz and smooth and claimed that these assumptions are standard in the related literature. This is not the case, most of the analysis in the literature actually make reasonable assumptions on the policy parametrization to prove these results.
> >
> > - Some typos make reading the new proof difficult (norms, parenthesis for squared terms, bars for conditioning …).
> > The assumptions to obtain the result are not clearly stated (definition of the Lipschitzness constant $L_{\psi}$ defined after being used, smoothness and Lipschitzness of $J$ …).
> > It is not clear how you obtain the inequality from the decomposition of the random PRE.
> > - Some notations are unclear, what is $u’$ in the decomposition of the PRE?
> > - In the quasi gradient norm bound, the last inequality is unclear as the replay buffer has a (random?) size depending on $t$ as well. It is unclear how it disappears. In the penultimate equality, the conditioning seems to be missing, it seems like a multiplicative $S(t+1)$.
> > - I think this result deserves a new complete round of review. I appreciate the authors’ efforts in improving their work but I think the paper requires significant rewriting, including writing down the precise assumptions and the obtained result with relevant discussions regarding its relevance and comparison to prior work.
> >
> > **Main concern:**
> >
> > - Given that prior work has already introduced zeroth order policy gradient methods, the positioning of the paper should be clear in this regard and should clearly state that this is an extension to actor-critic methods where the Q-function is approximated using function approximation. This should be clear from the introduction as this paper cannot claim novelty in this regard.
> >
> > -  Regarding Theorem 1, as I mentioned in my original review, the extension to add the critic error is a straightforward extension (essentially adding and subtracting the new critic term). In addition, little is said about this error or how to reduce it.
> >
> > - As per my initial review and as confirmed by the rebuttal, the main novelty seems to be the extension to actor-critic algorithms with deep neural network critic approximators and deep RL experiments on some benchmarks. I think this should be made clear in the paper.
> >
> > **Sample complexity:** In the rebuttal: ’Because our gradient estimator relies on random perturbations in the $p$-dimensional action space, its variance scales as $\mathcal{O}(p/N)$ , where $N$ is the batch size. Thus, obtaining an $\epsilon$-accurate estimate requires $N = \mathcal{O}(p/\epsilon^2)$ samples, reflecting linear dependence on $p$.’ If the variance if $\mathcal{O}(p/N)$, how do you obtain $N = \mathcal{O}(p/\epsilon^2)$? Is this result reported and proved in some part in the manuscript?
> >
> > **Q4:** I think it would be still useful to mention that the algorithm is not fully zeroth order as I mentioned in my initial review. Regarding RL applications where the zeroth order approach is the only possible approach for policy optimization, I was expecting more concrete applications. I think  it would be relevant to discuss these in the paper to strengthen motivation if there are any substantial ones.

---

> > > ### Author Response · Authors · 2025-11-12
> > > **Response to Reviewer's Follow-Up Comments**
> > >
> > > Thank you for your continued engagement in the discussion.
> > >
> > > ---
> > > ### **The first-order stationary guarantee for smooth non-convex optimization and the theoretical result in the prior work**
> > >
> > > To start, we are happy to cite and discuss the two papers mentioned by the reviewer in the revised version of the paper; this was already in our plans, and we apologize if this was not clear in our previous responses.
> > >
> > > While we appreciate the comments on the new convergence (or, rather, an ergodic stability) bound, we believe that the overall position towards the new result is overly critical. Our derivation is neither standard nor trivial, and we kindly welcome the reviewer to take a closer look at the proof, in particular at how the critic training is selected in a **posterior way** so that gradient incompatibility is (optimally) mitigated, as well as at the various intricacies related to sampling and maintaining valid stochastic gradients across the composition of the approximations imposed by the off-policy nature of the algorithm. These details are far from trivial, at least as compared with proving convergence of a conventional stochastic gradient method (even in the nonconvex setting). Our proof is also substantially different from those in the papers by Kumar and Vemula, roughly for the same reasons.
> > >
> > > To elaborate more, what is indeed standard in our development is the technique of exploiting smoothness to derive first-order stationary guarantees for nonconvex problems, as the reviewer correctly observes. However, we exploit this technique in order to study oCPG as an _inexact_ stochastic gradient method, where the inexactness stems from unavoidable errors that are induced by the presence of a _(i)_ replay buffer, _(ii)_ clipped double $Q$-learning, and _(iii)_ function approximation. Those errors, which are introduced by standard fashion in off-policy RL (as a means to approximate an otherwise completely unknown $Q$-function) are essentially irreducible, in the sense that they typically do not completely diminish during the operation of the algorithm. Therefore, one should not expect that gradient norms corresponding to the _true_ policy gradient converge to zero in any reasonable sense. Instead, one's best hope is that the off-policy algorithm (oCPG in particular) is **stable relative to those errors (in an appropriate "control theoretic" sense, so to say)**. This is precisely what we establish via the new result: **oCPG is (ergodically) stable to all induced gradient approximation errors**. Of course, stability to (arbitrary) gradient errors may not be guaranteed in general for other off-policy RL algorithms (such as TD3, DDPG, SAC, etc.), **but it does for oCPG** (by the way, the fact that the error terms depend on $T$ in our analysis does not imply anything other that the **errors are averaged out during the operation of the algorithm, which can  have a plausible effect in performance**; stepsizes also depend on $T$ by standard fashion).
> > >
> > > **Some additional remarks:**
> > > 1. Note that **it is the structure of oCPG that enables such analysis**. In fact, the analysis is **not** applicable to rivals such as TD3, mainly because of the inability to relate gradient errors to value-based $Q$-function approximation errors, which is an additional technical disadvantage of directly using function approximator gradients (on top of causing gradient incompatibility).
> > >
> > > 2. Our analysis reveals that the **compatibility error can indeed be controlled by proper critic training/fitting within the framework of oCPG** (while the remaining error terms remain irreducible as expected; see discussion above).
> > >
> > > ---
> > > ### **The most of the analysis in the literature _not_ assuming the objective is both Lipschitz and smooth**
> > >
> > > We would appreciate clarification on which analyses in the literature are being referred to. If the reviewer has specific references in mind that relax these assumptions or rely solely on structural restrictions of the policy parameterization, we would be glad to examine them and discuss how our analysis relates. Without explicit citations, it is unfortunately difficult to verify or respond further to this comment (for instance, Lipschitzness and smoothness are also impose in the paper by Kumar, which the reviewer has cited).
> > >
> > > ---
> > >
> > > ### **Typos in the analysis**
> > >
> > > Thanks for pointing those issues out; we will be updating the exposition of the proof accordingly. Additionally, the version of the proof appearing in the manuscript will be restructured in the form of clearly stated assumptions, preparatory lemmas, and a final theorem, resolving the concerns raised by the reviewer.
> > >
> > > ---
> > >
> > > ### **The notation $u^\prime$**
> > >
> > > This was a typo and it should have been the standard $u$. We have now fixed it—thanks for noticing.

---

> > > > ### Author Response · Authors · 2025-11-12
> > > > **Response to Reviewer's Follow-Up Comments (continued)**
> > > >
> > > > ### **The last inequality in the quasi-gradient norm bound**
> > > >
> > > > We clarify that $S(t+1)$ denotes the indices of the transitions in the minibatch sampled from the replay buffer, and $N$ is the corresponding batch size. There is no constraint or assumption on the total replay buffer size, and the analysis remains independent of it throughout.
> > > >
> > > >  The term $S(t+1)$ disappears because the standard multivariate Gaussian perturbations $\boldsymbol{u}_i$, sampled at the optimization step corresponding to the transition indexed by $i$, are independent of the state history (filtration), the replay buffer content, and the sampled indices. Therefore, $S(t+1)$ can be safely removed from the conditioning when integrating $\boldsymbol{u}$. We will make sure to explain this step in the manuscript version of the proof.
> > > >
> > > > Finally, the reduction to the last inequality follows from a standard relation derived in Nesterov & Spokoiny (2017).
> > > >
> > > > ---
> > > >
> > > > ### **Sample complexity**
> > > >
> > > > Thank you for catching this. The scaling statement in the rebuttal was a mistake. The variance of the two-point estimator actually scales as $\mathcal{O}(p^2/N)$, not $\mathcal{O}(p/N)$. Consequently, to achieve an $\varepsilon$-accurate estimate (in terms of variance), one requires $N = \mathcal{O}(p^2 / \varepsilon)$ samples. While this result is not stated it the paper, we can include it if the reviewer thinks it's important.
> > > >
> > > > ---
> > > >
> > > > ### **Q4**
> > > >
> > > > The algorithm is zeroth order **in the sense of policy optimization**, as it does not require action-value derivatives for the policy update. While the critic is trained using standard first-order TD learning—as in all actor–critic methods—this does not affect the zeroth-order nature of the policy update itself. That said, we acknowledge the first-order gradients computed in the overall algorithm.
> > > >
> > > > We will clarify this distinction in the revised version to prevent potential ambiguity.

---

> > > > > ### Comment · Reviewer_pnxs · 2025-11-14
> > > > > **Response**
> > > > >
> > > > > I thank the authors for their response. In my opinion, the paper is still not ready for publication. I maintain that this paper deserves significant rewriting to discuss prior work adequately, clarify the positioning of the paper, and report complete new results with their assumptions, relevant discussions and discuss new experiments. I believe this would lead to a significantly stronger paper. While I appreciate the efforts of the authors in adding new results and trying to clarify the technical novelty of the new proposed results, given all the planned additions to the manuscript (that cannot be immediately seen at this moment in a revised manuscript), the required modifications go far beyond a minor revision in my opinion. In addition to my previous comments, I would like to add a few elements and follow up on the responses of the authors with a few additional important concerns:
> > > > >
> > > > > **Claim of novelty.** The rebuttal claims that ‘this is the first convergence result for general off-policy RL on continuous state and action spaces’. This claim does not appear to be true, see e.g. the following paper about deterministic policy gradient (which is not cited nor discussed in the paper or in the rebuttal):
> > > > >
> > > > > Xiong et al. Deterministic Policy Gradient: Convergence Analysis. UAI 2022.
> > > > >
> > > > > While this result is different (as it uses a different compatible function approximation framework), a thorough literature review and discussion is needed.
> > > > >
> > > > > **Follow up on first-order stationarity.** The rebuttal proposed a new first-order stationarity result that was not initially present in the first submission after the first reviews which pointed out the work of Kumar et al. The rebuttal could have clearly mentioned that this result extends the existing one of Kumar et al. but this was not the case, which is rather confusing and concerning. I acknowledge that some of the errors cannot be reduced, this is the case of the function approximation error in general as I mention in my initial comment. Overall, the proof follows the same strategy as the previously established result of Kumar et al. which the paper did not initially discuss, despite a few additions. Note that the ‘quasi gradient norm bound’ also corresponds to Lemma 3 in Kumar et al. The proposed analysis does extend the analysis by considering a few additional errors by adding and substracting terms to make different errors show up in the final result and there are different sampling technicalities in the analysis (these are simply addressed using e.g.  independent sampling of the noise for zeroth order gradient which is independent from the replay buffer sampling for instance). Essentially, what is new here compared to prior work is to replace the zeroth-order true Q functions (induced by policies) by function approximation Q functions and then introducing the PRE as the difference between the estimated Q and its true value. The analysis then propagates these errors which remain as irreducible error floors in the final bound. This deserves to be mentioned and clarified.
> > > > >
> > > > > **About first-order stationarity for PG analysis.** The paper of Kumar et al. dates back to 2020. Since then, there were important advances in the analysis of policy gradient methods. Notably, it can be shown that policy gradient methods enjoy \textit{global} convergence rates despite the objective being nonconcave in the policy parameters. This is much stronger than first-order stationarity guarantees. This is thanks to a gradient domination proved a few years ago (see e.g. Agarwal et al. 2021, reference below). Despite these advances, the paper’s rebuttal focuses on first-order-stationarity guarantees, I think this should be justified per se. Can you show global convergence guarantees up to the errors that your method suffers? I think this would be a more up to date and interesting technical contribution. If the paper does not show this, it should at least discuss recent literature on the topic, justify why looking at first-order stationarity instead of aiming for stronger global optimality results which seem within reach. Overall even for first-order stationarity I believe the paper would benefit from incorporating a related work discussion and contrast the contributions for the zeroth order proposed algorithm with existing analysis and guarantees.
> > > > >
> > > > > Agarwal et al. On the Theory of Policy Gradient Methods: Optimality, Approximation, and Distribution Shift. JMLR 2021.
> > > > >
> > > > > Xiao. On the convergence rates of policy gradient methods. JMLR 2022.
> > > > >
> > > > > Fatkhullin et al. Stochastic Policy Gradient Methods: Improved Sample Complexity for Fisher-non-degenerate Policies. ICML 2023.

---

> ### Comment · Reviewer_pnxs · 2025-11-14
> **Response (2/2)**
>
> **Analysis of actor-critic methods with function approximation.** There are also a number of works analyzing actor-critic methods (even for continuous state action spaces), see e.g. references below. These works derive global convergence results rather than first-order stationary results. They are not considering zeroth order algorithms though and they rely on the standard compatible function approximation framework.
>
> Gaur et al. Closing the gap: Achieving global convergence (last iterate) of actor-critic under markovian sampling with neural network parametrization. ICML 2024.
>
> Ganesh et al. Order-Optimal Global Convergence for Actor-Critic with General Policy and Neural Critic Parametrization. UAI 2025.
>
> **Statements of assumptions.** Assumptions regarding function approximation and bias errors can be stated and clearly discussed, see some of the above works.
>
> **Lipschitzness and smoothness of the objective.** This is not a major point. Still, regarding references of papers showing the the assumptions are not standard as claimed by the rebuttal, please see below 5 different references for the continuous state action space setting which are not incorporated nor discussed in the paper (one of the reasons why the paper would require a thorough literature survey and rewriting, in addition to the previous elements):
>
> Pirotta et al. 2015 (Theorem 3), Papini et al. 2022 (section 3.3, Lemma 6, Theorem 7), Yuan et al. 2022 (Lemma 4.4), Xiong et al. 2022 (Lemma 1), Bedi et al. 2024 (Lemma 5.2).
>
> For Kumar et al. 2022, which was not mentioned in the original manuscript, you are right that the paper does suppose that the objective is smooth and Lipschitz continuous in the statement of their result (Theorem 2). However, note that they comment about this right after the theorem and mention that this can be easily satisfied under their previously made assumptions under adequate assumptions on the policy parameterization. I maintain though that it is not standard to make the assumption of smoothness if one can prove it under reasonable assumptions. While smoothness is a standard assumption in optimization in general, it is less so in RL where this objective has a particular structure which can be exploited to show that the objective does actually enjoy smoothness.
>
> Yuan et al. A general sample complexity analysis of vanilla policy gradient. AISTATS 2022.
>
> Papini et al. Smoothing policies and safe policy gradients. Machine Learning 2022.
>
> Xiong et al. Deterministic Policy Gradient: Convergence Analysis. UAI 2022.
>
> Pirotta et al. Policy gradient in Lipschitz Markov Decision Processes. Machine Learning 2015.
>
> Bedi et al. On the Sample Complexity and Metastability of Heavy-tailed Policy Search in Continuous Control. JMLR 2024.
>
> None of these references is discussed despite being relevant. The initial version of the paper did not have a first-order-stationarity guarantee though and was not focused on theory. I think there is some significant work to state reasonable assumptions clearly, state the result, discuss it compared to prior work adopting a zeroth order approach (Kumar et al. and others) and a first-order one as well (see references above).
>
> At this stage, I do not have any further questions.

---

### Review · Reviewer_uVQc · 2025-09-27

**Summary Of Contributions:**

The paper introduces Off-Policy Compatible Policy Gradient (OCPG), an actor-critic algorithm that uses a zeroth-order approximation via two-point stochastic estimation to compute the action-value gradient. This method provably addresses the incompatibility issues in deterministic policy gradient (DPG) algorithms using deep function approximation. Empirically, OCPG frequently matches or exceeds the performance of state-of-the-art methods.

**Audience:**

Yes

**Audience Explanation:**

See **Strengths**.

**Broader Impact Concerns:**

None.

**Claims And Evidence:**

Yes

**Claims Explanation:**

Strengths:
1. The introduction of two-point stochastic gradient estimation within the action space to approximate $\nabla_a Q$ is a novel, elegant solution that entirely bypasses the need for the unreliable first-order gradient computation in DPG/TD3, fundamentally addressing the long-standing compatibility problem.
2. The framework includes Theorem 1, which provides a concrete and provable $\epsilon$-compatibility guarantee on the approximation error $||\hat{\nabla}^{\mu, \psi} J(\theta)-\nabla J(\theta)||$.
3. The approach generalizes the classical compatibility conditions, which strictly required Q-function gradients to be linear in their parameters, thus removing a severe restriction and enabling the sound use of highly expressive deep neural networks as critics.

**Requested Changes:**

Weaknesses:
1. The theorem gives $\epsilon$-compatibility under a generic state distribution, but the algorithm is trained off policy from a replay buffer and the paper doesn't analyze the extra errors this introduces. In particular, distribution mismatch, interaction with two-point perturbations (importance weighting), and bootstrapping bias aren't decomposed, so it's unclear how the bound translates to the practical off-policy setting.
2. The optimal tuning of the smoothing parameter $\mu$ is difficult as it relies on balancing the Perturbation Representation Error (PRE) term with the smoothing error term, requiring knowledge of the unknown Lipschitz constant $G$ of the true Q-function $Q^{\pi_\theta}$.

Questions:
1. Since $\mathbb{E}_u[\frac{Q(s, a)}{\mu} u]=0$, what is the specific variance reduction gained by retaining this term as an implicit baseline compared to a purely onesided zeroth-order estimator?
2. How would an importance-sampling correction mechanism be implemented for the CPG estimator $\hat{\nabla}^{\mu, \psi} J(\theta)$, given that the $Q$ evaluations are performed on the perturbed actions $a+\mu u$ ?
3. Would directly minimizing the Perturbation Representation Error (PRE) $\epsilon_{\mu, \psi}^{\pi_\theta}$, with respect to $\psi$ (perhaps as an auxiliary loss) be more effective for ensuring $\epsilon$-compatibility than relying solely on the standard TD-error minimization?
4. Does the $\epsilon$-compatibility result of Theorem 1 fundamentally rely on the Gaussian distribution for the perturbation $u \sim \mathcal{N}\left(0, I_p\right)$, or could similar bounds be proven for other, potentially lower-variance, smoothing distributions (e.g., Uniform over a sphere)?

---

> ### Author Response · Authors · 2025-10-20
> **Response**
>
> Thank you for your positive assessment and constructive questions.
>
> ---
>
> ### **Question 1**
>
> Let us show the variance reduction explicitly. Consider the one-sided estimator without baseline
>
> $
> g\_\text{no-base} = \frac{Q(s,a+\mu u)}{\mu} u,
> $
>
> and the same estimator with the (implicit) baseline
>
> $
> g\_\text{base} = \frac{Q(s,a+\mu u)-Q(s,a)}{\mu} u,
> $
>
> where $u \sim \mathcal{N}(0, I\_p)$. Since $\mathbb{E}[u]=0$, both have the same mean:
> $\mathbb{E}[g\_\text{no-base}]=\mathbb{E}[g\_\text{base}]=\nabla\_a Q\_\mu(s,a)$.
>
> Let $q\_0:=Q(s,a)$ and $g:=\nabla\_a Q(s,a)$. For small $\mu$,
>
> $
> Q(s,a+\mu u) = q\_0 + \mu g^\top u + o(\mu),
> $
>
> Dividing both sides by $\mu$ and multiplying by $u$, we have
>
> $g\_\text{no-base} = \frac{q\_0}{\mu} u + (g^\top u)u + o(1), \qquad g\_\text{base} = (g^\top u)u + o(1).$
>
> Therefore, up to $o(1)$ terms,
>
> $
> \mathbb{E}\big[\|g\_\text{no-base}\|^2\big] = \frac{q\_0^2}{\mu^2} \mathbb{E}\left[\|u\|^2\right] + 2\frac{q\_0}{\mu} \mathbb{E}\left[(g^\top u)\|u\|^2\right] + \mathbb{E}\left[(g^\top u)^2\|u\|^2\right],
> $
>
> $
> \mathbb{E}\big[\|g\_\text{base}\|^2\big] = \mathbb{E}\left[(g^\top u)^2\|u\|^2\right].
> $
>
> The mixed term vanishes by symmetry (odd moments of standard Gaussian is zero):
> $\mathbb{E}[(g^\top u)\|u\|^2]=0$.
> Using standard Gaussian identities, we get
> $\mathbb{E}[(g^\top u)^2\|u\|^2]=\|g\|^2 (p+2)$ and $\mathbb{E}[\|u\|^2]=p$.
> Hence,
>
> $
> \mathbb{E}\big[\|g\_\text{no-base}\|^2\big] = \frac{q\_0^2}{\mu^2} p + \|g\|^2 (p+2) + o(1), \qquad \mathbb{E}\big[\|g\_\text{base}\|^2\big] = \|g\|^2 (p+2) + o(1).
> $
>
> To first order in $\mu$,
>
> $
> \mathrm{Var}(g\_\text{no-base})-\mathrm{Var}(g\_\text{base}) = \frac{Q(s,a)^2}{\mu^2} p + o\left(\frac{1}{\mu^2}\right),
> $
>
> i.e., retaining the $- Q(s,a)$ baseline removes the dominant
> $\Theta\left(\tfrac{1}{\mu^2}\right)$ variance term coming from the constant component of $Q(s,a+\mu u)$, while leaving the mean unchanged. The reduction scales with the action dimension $p$.
>
> ---
> ### **Question 2**
>
> We would like to clarify that our work does not study off-policy correction mechanisms in detail; rather, the proposed estimator is designed to be compatible with an off-policy framework in the sense of actor–critic training with replay buffers. Since we work around the $Q$-function—already central to standard off-policy methods—our design naturally adopts the off-policy perspective.
>
> In principle, one could adapt standard off-policy corrections by weighting the perturbed action evaluations with ratios between the behavior distribution and the perturbed distribution. Importance sampling methods assume a stochastic policy, which in our setting can be realized through the Gaussian noise–injected exploration actions, while deterministic evaluations are still used for optimization. Techniques such as Retrace or V-trace [1, 2] could then be extended by including $\tfrac{\pi(a+\mu u \mid s)}{\beta(a+\mu u \mid s)}$ as a multiplicative correction factor. While this is left for future work in our study, it provides a straightforward direction for integrating importance sampling corrections if one wishes to control for discrepancies between the perturbation distribution and the behavior policy.

---

> ### Author Response · Authors · 2025-10-20
> **Response (continued)**
>
> ### **Question 3**
>
> Thank you for this insightful question. PRE is a value-space mismatch at *perturbed* actions:
>
> $
> \varepsilon^{\pi\_\theta}\_{\mu,\psi} = \sqrt{\mathbb{E}\_{s,u}\big[ \big\|Q\_\psi\big(s,\pi\_\theta(s)+\mu u\big)-Q^{\pi\_\theta}\big(s,\pi\_\theta(s)+\mu u\big)\big\|^2\big]}.
> $
>
> In our implementation (Algorithm 1), the critic is trained with Clipped Double Q-learning and *target policy smoothing*, i.e.,
>
> $
> \mathcal{L}\_{\text{TD}}(\psi) =\mathbb{E}\Big[\big(y - Q\_\psi(s,a)\big)^2\Big],\quad \text{where} \quad y=r+\gamma \min\_{i=1,2} Q\_{\psi\_i'}\big(s', \pi\_{\theta'}(s')+\epsilon\big),
> $
>
> with $\epsilon$ clipped Gaussian noise (note that this is not related to smoothing; it comes from Clipped Double Q-learning). This training implicitly minimizes $\varepsilon^{\pi\_\theta}\_{\mu,\psi}$ for two reasons:
>
> (i) _**Bellman consistency on perturbed actions.**_ The target uses next-actions of the form $\pi\_{\theta'}(s')+\epsilon$, placing supervised signal precisely on the same kind of *action perturbations* that appear in our gradient surrogate ($\pi\_\theta(s)+\mu u$). As the critic fits these targets across replayed transitions (which already cover noisy/perturbed actions), the value approximation error at perturbed actions shrinks, thereby decreasing PRE.
>
> (ii) _**Distributional alignment with the CPG estimator.**_ The actor gradient uses two-point differences at $a=\pi\_\theta(s)$ and $a+\mu u$. Training the critic on (and around) these neighborhoods via policy-smoothing targets aligns the critic's accuracy with where the CPG estimator queries it, directly improving the bound in Theorem~1.
>
> Overall, yes—our training already *effectively* minimizes PRE by fitting $Q\_\psi$ on perturbed actions via TD with target policy smoothing (as part of Clipped Double Q-learning) which is exactly where PRE is defined. An explicit auxiliary term like $\mathcal{L}\_{\text{PRE-TD}}$ is a drop-in, principled variant; it may tighten compatibility in practice but was not necessary for the reported results.
>
> ---
>
> ### **Question 4**
>
> The $\epsilon$-compatibility bound in Theorem 1 does not fundamentally rely on the perturbation being Gaussian — only on a few generic properties:
>
> 1. Zero mean and isotropy (or unbiasedness): $\mathbb{E}[u] = 0$, and $\mathbb{E}[u u^\top] \propto I$.
> 2. Finite second moments/bounded variance.
> 3. Smoothness-based Taylor expansion control so that the action-gradient linearization error is small for small $\mu$, what we have used in the derivation in the response to the first question.
>
> Under these mild assumptions, one can replicate the steps in the proof (expanding $Q^{\pi\_\theta}(s, \pi\_\theta(s)+\mu u)$, decomposing bias, and bounding variance) and derive an analogous bound with the same scaling in $\mu$, dimension $p$, and PRE. The key constants (e.g. variance of $u$) may differ, but the qualitative form holds.
>
> That said, Gaussian perturbations often lead to the "cleanest" constants and tightest bounds in smoothing analyses (via Stein's identity and isotropic tail control). For example, many zeroth-order optimization works (e.g., Nesterov & Spokoiny [3], and follow-ups) prefer Gaussian smoothing precisely because it minimizes estimator variance under certain constraints (e.g., in "minimum-variance two-point estimators").
>
> **In short:** Yes, our result extends beyond Gaussian under standard regularity, but Gaussian is analytically convenient.
>
> ---
>
> ### **References**
>
> [1] Remi Munos, Tom Stepleton, Anna Harutyunyan, and Marc Bellemare. _Safe and efficient off-policy reinforcement learning._ NeurIPS, 2016.
>
> [2] Lasse Espeholt, Hubert Soyer, Remi Munos, Karen Simonyan, Vlad Mnih, Tom Ward, Yotam Doron, Vlad Firoiu, Tim Harley, Iain Dunning, Shane Legg, and Koray Kavukcuoglu. _IMPALA: Scalable distributed deep-RL with importance weighted actor-learner architectures._ ICML, 2018.
>
> [3] Yurii Nesterov and Vladimir Spokoiny. _Random gradient-free minimization of convex functions._ Foundations of Computational Mathematics.

---

### Review · Reviewer_7Y1W · 2025-10-06

**Summary Of Contributions:**

This paper proposes a new actor-critic algorithm for deterministic policy optimization in continuous control. The method replaces the gradient of the Q-function with a zeroth-order approximation obtained via Gaussian smoothing (applying a scaled Gaussian noise to the action), allowing policy updates based on two-point stochastic gradient estimation in the action space. The authors provide a theoretical upper bound on the difference between the true and approximated policy gradients and present empirical results demonstrating that their off-policy variant (oCPG) matches or outperforms strong baselines such as TD3 and SAC on MuJoCo benchmark tasks.

### Strengths:

- The motivation of instability and incompatibility of Q-function gradients in DPG is clear and well-founded.
- The theoretical section is well structured, with proofs that connect the smoothed Q-function to a provably bounded approximation of the true gradient.
- Experimental results show consistent performance gains or parity with strong baselines across canonical tasks.
- Ablation studies (on sparse, delayed, and noisy rewards) add useful insights into robustness.

### Weaknesses:

- The theoretical bound in Theorem 1 appears to be very loose. This makes it questionable how the claimed “arbitrarily small approximation error” can actually be achieved.
- The link between gradient approximation and policy convergence guarantees is not analyzed.
- The experimental comparisons are potentially unfair, as μ is tuned for oCPG but reused for TD3, and PPO hyperparameters and drastically poor performance are not discussed.
- The paper focuses solely on sample efficiency metrics, omitting diagnostic analyses of gradient or value accuracy that would strengthen the argument.

**Audience:**

Yes

**Audience Explanation:**

The paper addresses an important and recognized weakness in deterministic policy gradient methods: gradient incompatibility under non-linear function approximation. Its proposal of a zeroth-order policy gradient that maintains compatibility without explicit gradient computation is novel and potentially impactful. Even though theoretical and experimental aspects could be strengthened, the idea is timely and will likely interest researchers in reinforcement learning theory and algorithm design, particularly those exploring gradient-free or stable actor-critic updates.

**Broader Impact Concerns:**

No concerns.

**Claims And Evidence:**

No

**Claims Explanation:**

- The main theoretical claim that the gradient approximation error can be controlled at will is not well supported, since the bound includes the constant B that make it effectively uninformative. Precisely, the bound scales with B (even when $\upvarepsilon^{\pi_\theta}_{\mu,\uppsi}=0$}), and B is at least as large as the largest possible policy gradient.

- The paper does not demonstrate that the approximated gradients correlate with true or compatible policy gradients. They only prove ϵ-Compatible Policy Gradient (which is a new notion introduced by the paper). The gradient approximation may be close to the true one in Euclidean norm, but the gradient itself can still be pointing in opposite directions (which is one of the main problems this approximation was meant to solve). There are no diagnostic experiments or ablations to probe the quality of learned critics or gradients.

- Experimental fairness is questionable. The authors hyperparameter tune μ for oCPG, and then use that same value for TD3. This doesn't seem fair to the TD3 baseline and seriously puts into question the comparison results, especially given that the authors observe that μ < 0.1 performs poorly for their approach. It's also unclear why PPO has relatively poor performance in many of the tasks, even failing drastically in the Ant and Humanoid ones. This performance gap was not discussed and the hyperparameters for PPO where not reported. PPO is also not cited in Section 5.

Thus, while the algorithm is promising, the evidence does not convincingly support all theoretical or empirical claims.

**Requested Changes:**

### Major

- Clarify and tighten Theorem 1’s bound.
  - Address the looseness of the upper bound in Theorem 1. Mainly, can you explain how B affects practical tightness?
  - Revise or justify the claim that the gradient approximation error can be “controlled at will” and "made arbitrarily small".
  - Discuss the implications of the gradient approximation on convergence guarantees and directionality of policy gradients.

- Improve experimental rigor:
  - Re-tune the hyperparameter μ for TD3 to ensure fairness.
  - Provide full details on PPO hyperparameters and explain its poor performance in most tasks (especially Ant and Humanoid).
  - Include additional analyses or metrics examining the quality of learned value functions (and possibly gradient estimates), not just sample efficiency. This would empirically support claims about compatibility and gradient fidelity.

### Minor

- Inconsistent use of bold for vectors throughout. E.g. In equation 5 u is a vector (bold), but are s and a not also vectors (they are not bolded)?
- Avoid abbreviations like i.e. and e.g., and avoid in formal language like "In any case,"
- Omitting $1-\gamma$ (Section 5.1.) in the implementation for "simplicity" is not a sufficient justification. Maybe say it is because it doesn't affect the gradient direction?
- $\tau$ and c are undefined in Algorithm 1

---

> ### Author Response · Authors · 2025-10-20
> **Response — Part I**
>
> Thank you for your detailed and thoughtful feedback, and the recognition of the paper’s motivation.
>
> ---
>
> ## Bound of Theorem 1
>
> ### **The $B$ Term**
>
> We agree that the bound in Theorem 1 can be numerically loose (including the newly added convergence analysis; see our subsequent responses). The factor $B$ is the supremum of the policy Jacobian norm, i.e., $B \triangleq \sup\_{(s,\theta)}\|\nabla\_\theta \pi\_\theta(s)\|$, and it primarily reflects the actor's smoothness. In practice, $B$ is constrained by architectural choices (e.g., bounded activations such as $\tanh$), normalization, and standard regularization (weight decay, spectral norm control, gradient clipping), so it behaves as a stable multiplicative constant rather than a tunable parameter. Its role is to scale the error bound with respect to the actor's sensitivity; it does not, by itself, determine optimization quality.
>
> Importantly, the purpose of the result is to *expose dependencies*—on $\mu$, the action dimension $p$, the PRE, $\gamma$, and the actor smoothness via $B$—to *inform practical implementations*, not to provide the exact/tightest constants (which are often unattainable or problem-specific). From this perspective, $B$ offers a clear interpretation of how actor smoothness affects the approximation error, even if the constant is not tight in a numerical sense.
>
> ---
>
> ### **Claims:** _“Controlled at will”_ and _"made arbitrarily small"_
>
> The phrasing "controlled at will" and "made arbitrarily small" was intended in a theoretical—not necessarily strictly practical—sense. Theorem 1 implies that the gradient approximation error depends on two terms: the PRE and the smoothing scale $\mu$. In principle, if the critic perfectly minimizes the PRE and $\mu$ approaches zero, the approximation error can be made arbitrarily small. While such overfitting or infinitesimal $\mu$ are unattainable in practice, this formulation highlights that both factors are **explicitly controllable** and their contributions to the total error are quantifiable. Hence, the result serves to demonstrate that the approximation accuracy is governed by tunable and measurable quantities, rather than asserting that the bound can be achieved exactly in implementation.
>
> ---
>
> ### **Convergence Analysis**
>
> We are currently finalizing a detailed convergence analysis of oCPG. While the proof is not yet complete, we are posting our responses now to meet the rebuttal timeline. We prefer to take the necessary time to ensure the analysis meets full theoretical rigor rather than providing a preliminary version. We will post the complete proof during the week of October 20.
>
> The convergence analysis explicitly incorporates the effects of the replay buffer (addressing off-policy discrepancies) and the Q-network approximation through the Clipped Double Q-learning algorithm—an integration that, to the best of our knowledge, **has not been formally analyzed in the literature before**.
>
> We hope the reviewer is fine with this timeline.
>
> ---
>
> ## Experimental Rigor
>
> ### **Re-tuning $\mu$ for Fairness**
>
> We tuned the parameter $\mu$ and found the optimal values to be $\mu=0.1, 0.125$. For all reported comparisons, we deliberately fixed $\mu=0.1$ to align with the original TD3 setting (as noted in the second column of page 7 in [2]). This ensured fairness across methods by using the canonical TD3 configuration.
>
> The choice and its rationale were described in the initial submission (see Section 5.1, paragraph "Selection of $\mu$"), and remain unchanged in the current version:
> > To choose between 0.1 and 0.125, we aligned $\mu$ with the standard deviation of the Gaussian exploration noise used in TD3, which is 0.1. This alignment ensures *(i)* fair evaluations with TD3, *(ii)* an adequate exploration, and *(iii)* a moderate value for smoothing. Notably, our method does not introduce any additional hyperparameters, as compared to the current state-of-the-art.
>
> ---
>
> ### **PPO Hyperparameters**
>
> Thank you for pointing out these.
>
> **Hyperparameters and Implementation.** We used the default PPO setup (including neural network architecture and generalized advantage estimation) from the original paper, with implementation based on the widely used [repository](https://github.com/ikostrikov/pytorch-a2c-ppo-acktr-gail) (3.8k GitHub stars). We initialized with the repository's default hyperparameters and made minor adjustments informed by our prior deep RL projects.
>
> *(Continues below)*

---

> ### Author Response · Authors · 2025-10-20
> **Response — Part II**
>
> ### **PPO Hyperparameters** _(continued)_
>
> **Poor Performance in Ant and Humanoid.** Since it is an on-policy method and relatively sample-inefficient in high dimensions, PPO does not converge within our 1M–2M step budgets in Ant and Humanoid, where the state–action dimensionalities are (105, 8) and (348, 17), compared to much smaller ones in Walker2d (17, 6) and Hopper (11, 3). Prior work (e.g., [1]) shows that PPO typically requires on the order of 10M+ steps to achieve strong locomotion performance in these tasks. In our experiments, PPO was included as a reference point; our primary comparisons are with off-policy methods (TD3, SAC, oCPG), where sample efficiency and stability are the more appropriate metrics.
>
> We have added the PPO hyperparameter and implementation details to Appendix B.1, and the explanation for Ant and Humanoid performance as a paragraph in Section 5.2.
>
> ---
>
> ### **Analysis of Learned Value Functions and Gradient Estimates**
>
> Thank you for the helpful suggestion. To study the quality of gradient estimates, we performed a controlled diagnostic study in a linear quadratic regulator (LQR) setting, where exact Q-values and policy gradients are analytically available.
>
> _**Setup.**_ We consider a discrete-time LQR with dynamics
>
> $
> s\_{t+1} = A s\_t + B a\_t + w\_t,\quad w\_t \sim \mathcal{N}(0,0.1^2I),
> $
>
> and quadratic costs
>
> $
> c(s,a) = s^\top Q\_c s + a^\top R\_c a,\quad Q\_c = I,\ R\_c = \mathrm{diag}(0.01,0.1,1,0.01,0.1,1).
> $
>
> The optimal solution is obtained via the discrete algebraic Riccati equation, giving access to the exact Q-function and its gradients. We adopt a linear policy $\pi\_\theta(s) = K\_\theta s$ and construct a static dataset of 500 state–action pairs from a behavior policy $a = 0.7K^* s$, where $K^*$ is the optimal feedback gain. This ensures coverage of near-optimal actions while retaining sufficient exploration. To provide local information, we also include perturbed neighbors $(a \pm \mu u)$ with $\mu=0.005$ and $u$ uniformly sampled from the unit sphere. The critic is a two-layer ReLU MLP (256 units per layer) trained by regression on the true Q-values, following standard actor–critic practice, e.g., in TD3 and SAC.
>
> _**Gradient Estimators.**_ We consider:
>
> - the true gradient $\nabla J(\theta) = \frac{1}{N}\sum\_{i=1}^N\nabla \pi(s\_i)\nabla\_a Q^*(s\_i, a\_i)\vert\_{a\_i = \pi\_\theta(s\_i)}$,
> - the DPG estimate $\nabla \hat{J}\_\text{DPG}(\theta) = \frac{1}{N}\sum\_{i=1}^N \nabla \pi(s\_i) \nabla Q\_\psi(s\_i, a\_i)\vert\_{a\_i = \pi\_\theta(s\_i)}$,
> - and the symmetric zeroth-order estimator
>
> $
> \hat{J}\_{\text{ZOO}}(\theta)= \frac{1}{N}\sum\_{i=1}^N \nabla \pi(s\_i) \left.\left[ \frac{Q\_\psi(s\_i, a\_i+\mu u\_i) - Q\_\psi(s\_i, a\_i - \mu u\_i)}{2\mu} u\_i \right]\right|\_{a\_i=\pi\_\theta(s\_i)},
> $
>
> where $N$ is the batch size, and compare:
>
> - Relative gradient estimation error of DPG: $\frac{\|\nabla J(\theta) - \nabla \hat{J}\_\text{DPG}(\theta)\|}{\|\nabla J(\theta)\|}$
> - Relative gradient estimation error of CPT: $\frac{\|\nabla J(\theta) - \nabla \hat{J}\_\text{ZOO}(\theta)\|}{\|\nabla J(\theta)\|}$
>
> _**Findings.**_
> We include the Q-value and gradient error curves in the revised paper and make them available in our *anonymous* [GitHub repository](https://anonymous.4open.science/r/compat-grad-approx/figs/lqr_example/gradient_errors.png).
>
> We find that the critic learns accurate Q-values with negligible error. However, gradient accuracy differs markedly: the DPG estimate exhibits growing error (relative norm rising from $\sim$1.0 to $\sim$2.4), reflecting drift in critic derivatives, while the zeroth-order estimator remains consistently near 1.0, preserving alignment with the true gradient.
>
> _**Implications.**_ Even when value estimates are highly accurate, analytic gradients of a ReLU critic can drift substantially due to nonsmoothness and finite-sample effects. Our estimator avoids this issue by relying only on critic evaluations at perturbed actions. In the LQR study, this was ensured by explicitly adding perturbed samples to the dataset.
>
> Note that in off-policy settings, (e.g., main experiments in the paper), the same effect arises naturally because Gaussian exploration noise generates perturbed actions (with the same $\mu$); the replay buffer therefore already contains the required distribution of samples, ensuring that the critic is trained on exactly the inputs needed for our estimator.
>
> We now report this diagnostic study as Section 5.2 ("Linear Quadratic Regulator Example"), which complements the main evaluation results and provides empirical evidence for the gradient fidelity and compatibility of our approach.
>
> ---
>
> ## Minor
>
> Thank you for noticing these. We have now fixed them.

---

> > ### Author Response · Authors · 2025-10-20
> > **References**
> >
> > [1] Tuomas Haarnoja, Aurick Zhou, Pieter Abbeel, and Sergey Levine. _Soft actor-critic: Off-policy maximum entropy deep reinforcement learning with a stochastic actor._ ICML, 2018.
> >
> > [2] Scott Fujimoto, Herke Hoof, and David Meger. _Addressing function approximation error in actor-critic methods._ ICML, 2018.

---

### Author Response · Authors · 2025-10-20
**General Response**

We thank the Action Editor and all reviewers for their constructive and insightful feedback. We are encouraged that the motivation, theoretical framing, and relevance of our work were recognized as valuable to the TMLR audience.

Our key revisions and additions are summarized below:

- **Theoretical framing:** We refined the motivation and clarified the implications of our theoretical results. This has improved the interpretation of Theorem 1 and its practical meaning.

- **Relation to prior work:** We expanded the Related Work section to clarify how our formulation extends prior studies on zeroth-order optimization. Specifically, we highlight the incorporation of a learned critic and the analysis of the induced bias through the newly introduced _perturbation representation error_ (PRE).

- **Sample complexity and limitations:** A new paragraph in Section 4.2.1 discusses the sample complexity of our estimator and the known dimensional dependence of zeroth-order methods.

- **Rigor in presentation:** We provided complete hyperparameter details, clarified implementation choices, and improved the technical preliminaries for precision and consistency.

---
### **New Experiment:** Gradient Estimation Analysis
We added a controlled linear quadratic regulator (LQR) experiment where exact policy gradients are available. This setup enables a direct comparison between DPG and our proposed estimator.

Results show that our method maintains stable and accurate gradient estimates, whereas DPG suffers from growing deviation despite accurate value fitting—empirically validating our theoretical findings.

---
### **Forthcoming Analysis:** Convergence Guarantee
We are completing a rigorous convergence analysis of oCPG that incorporates both the replay buffer (off-policy discrepancy) and Q-network approximation through the Clipped Double Q-learning algorithm. To our knowledge, such an integrated treatment has not appeared in prior work.

While the proof is not yet finalized, we are submitting our responses to meet the rebuttal schedule and will post the full analysis during the week of October 20. We prefer to ensure theoretical rigor rather than release a preliminary version.

---

All feasible revisions have been incorporated into the updated manuscript, and the paper will be further revised once the convergence proof is complete. We believe these updates substantially strengthen both the theoretical and empirical foundations of our work.

---

### Author Response · Authors · 2025-10-29
**Convergence Analysis — Part I**

We apologize for the slight delay in providing this analysis. We wanted to ensure the mathematical derivations were thoroughly verified and clearly presented, as this represents a significant theoretical contribution that required careful preparation.

---

### **Convergence Analysis**

We are able to provide an analysis of oCPG as an *inexact stochastic gradient method*, where the inexactness stems from the use of $Q$-function approximation (through double clipped $Q$-learning), the presence of a replay buffer, as well as the issue of gradient incompatibility (which we address specifically in our work).

We would like to emphasize that **the analysis is enabled particularly due to the use of (Gaussian) smoothing in our setting**. Therefore, our zeroth-order approach not only exhibits superior empirical performance, but also enables rigorous convergence analysis of the resulting algorithm (oCPG).

To the best of our knowledge, *this is the first convergence result for general off-policy RL on continuous state and action spaces* (if the AE/reviewers know of related references that we are not aware of, we will be happy to look at them and discuss them). The steps of the analysis are as follows; in the updated manuscript, we will include the analysis in the form of preliminary lemmata, and a final convergence theorem.

***Quasigradient Error Analysis:***

Let us first recall the oCPG gradient update, reading

$
\hat{\nabla}\_{\beta\_{t}}^{\mu,\psi\_{t+1}}J(\theta\_{t})=\dfrac{1}{(1-\gamma)N}\sum\_{i\in S(t+1)}\nabla\_{\theta}\pi\_{\theta\_{t}}(s\_{i})\dfrac{Q\_{\psi\_{t+1}}(s\_{i},a+\mu\boldsymbol{u}\_{i})-Q\_{\psi\_{t+1}}(s\_{i},a)}{\mu}\boldsymbol{u}\_{i}\Bigg|\_{a=\pi\_{\theta\_{t}}(s\_{i})},
$

where we have added time/iteration subscripts for the purpose of our analysis, and where $S(t+1)$ denotes the random set of indices drawn uniformly without replacement from the (also random) replay buffer ${\cal B}\_{t+1}$; i.e., at each $t$, $S(t)\subset[N\_{t}]$, where $N\_{t}$ denotes the total length of the replay buffer at iteration $t$. Together with the oCPG update, we also define the *off-policy* quasi-gradient relying on *exact* $Q$-values as

$
\nabla\_{\beta\_{t+1}}J(\theta\_{t})=\dfrac{1}{(1-\gamma)N\_{t}}\sum\_{i=1}^{N\_{t}}\nabla\_{\theta}\pi\_{\theta\_{t}}(s\_{i})\nabla\_{a}Q^{\pi\_{\theta\_{t}}}(s\_{i},a)\big]\big|\_{a=\pi\_{\theta\_{t}}(s\_{i})}.
$

Note that while this quantity is not available in practice, we will employ it here as a "ghost proxy" in the convergence analysis of oCPG.

We can formulate the instantaneous gradient error decomposition

$
\hat{\nabla}\_{\beta\_{t+1}}^{\mu,\psi\_{t+1}}J(\theta\_{t})-\nabla J(\theta\_{t}) = \underbrace{\hat{\nabla}\_{\beta\_{t+1}}^{\mu,\psi\_{t+1}}J(\theta\_{t})-\nabla\_{\beta\_{t+1}}J(\theta\_{t})}\_{\text{PRE + Smoothing Error}}+\underbrace{\nabla\_{\beta\_{t+1}}J(\theta\_{t})-\nabla J(\theta\_{t})}\_{\text{Weak Replay Error}}.
$

Let us work with the first term on the right-hand side above. To this end, let $\{\mathscr{F}\_{t}\}\_{t}$ be the filtration generated by all the information (random variables) that is realized up to and including iteration $t$ of oCPG. We have, for fixed critic choices $\psi\_{t+1}$ measurable relative to $\mathscr{F}\_{t},{\cal B}\_{t+1}$ for all $t$,

$
(1-\gamma)\mathbb{E}\big[\hat{\nabla}\_{\beta\_{t+1}}^{\mu,\psi\_{t+1}}J(\theta\_{t})\big|\mathscr{F}\_{t}\big] = \dfrac{1}{N}\mathbb{E}\Bigg[\sum\_{i\in S(t+1)}\nabla\_{\theta}\pi\_{\theta\_{t}}(s\_{i})\dfrac{Q\_{\psi\_{t+1}}(s\_{i},a+\mu\boldsymbol{u}\_{i})-Q\_{\psi\_{t+1}}(s\_{i},a)}{\mu}\boldsymbol{u}\_{i}\Bigg|\_{a=\pi\_{\theta\_{t}}(s\_{i})}\Bigg|\mathscr{F}\_{t}\Bigg]
$

$
= \dfrac{1}{N}\mathbb{E}\Bigg[\mathbb{E}\Bigg[\sum\_{i\in S(t+1)}\nabla\_{\theta}\pi\_{\theta\_{t}}(s\_{i})\dfrac{Q\_{\psi\_{t+1}}(s\_{i},a+\mu\boldsymbol{u}\_{i})-Q\_{\psi\_{t+1}}(s\_{i},a)}{\mu}\boldsymbol{u}\_{i}\Bigg|\_{a=\pi\_{\theta\_{t}}(s\_{i})}\Bigg|\mathscr{F}\_{t},S(t+1),{\cal B}\_{t+1}\Bigg]\Bigg|\mathscr{F}\_{t}\Bigg]
$

$
= \dfrac{1}{N}\mathbb{E}\Bigg[\sum\_{i\in S(t+1)}\nabla\_{\theta}\pi\_{\theta\_{t}}(s\_{i})\mathbb{E}\Bigg[\dfrac{Q\_{\psi\_{t+1}}(s\_{i},a+\mu\boldsymbol{u}\_{i})-Q\_{\psi\_{t+1}}(s\_{i},a)}{\mu}\boldsymbol{u}\_{i}\Bigg|\mathscr{F}\_{t},S(t+1),{\cal B}\_{t+1}\Bigg]\Bigg|\_{a=\pi\_{\theta\_{t}}(s\_{i})}\Bigg|\mathscr{F}\_{t}\Bigg]
$

$
= \dfrac{1}{N}\mathbb{E}\Bigg[\sum\_{i\in S(t+1)}\nabla\_{\theta}\pi\_{\theta\_{t}}(s\_{i})\mathbb{E}\_{\boldsymbol{u}\sim{\cal N}(0,\boldsymbol{I})}\Bigg[\dfrac{Q\_{\psi\_{t+1}}(s\_{i},a+\mu\boldsymbol{u})-Q\_{\psi\_{t+1}}(s\_{i},a)}{\mu}\boldsymbol{u}\Bigg]\Bigg|\_{a=\pi\_{\theta\_{t}}(s\_{i})}\Bigg|\mathscr{F}\_{t}\Bigg]
$

---

> ### Author Response · Authors · 2025-10-29
> **Convergence Analysis — Part II**
>
> $
> = \mathbb{E}\Bigg[\mathbb{E}\Bigg[\dfrac{1}{N}\sum\_{i\in S(t+1)}\nabla\_{\theta}\pi\_{\theta\_{t}}(s\_{i})\dfrac{Q\_{\psi\_{t+1}}(s\_{i},a+\mu\boldsymbol{u}\_{i})-Q\_{\psi\_{t+1}}(s\_{i},a)}{\mu}\boldsymbol{u}\_{i}\Bigg|\_{a=\pi\_{\theta\_{t}}(s\_{i})}\Bigg|\mathscr{F}\_{t},{\cal B}\_{t+1}\Bigg]\Bigg|\mathscr{F}\_{t}\Bigg]
> $
>
> $
> = \mathbb{E}\Bigg[\dfrac{1}{N\_{t}}\sum\_{i=1}^{N\_{t}}\nabla\_{\theta}\pi\_{\theta\_{t}}(s\_{i})\mathbb{E}\_{\boldsymbol{u}\sim{\cal N}(0,\boldsymbol{I})}\Bigg[\dfrac{Q\_{\psi\_{t+1}}(s\_{i},a+\mu\boldsymbol{u})-Q\_{\psi\_{t+1}}(s\_{i},a)}{\mu}\boldsymbol{u}\Bigg]\Bigg|\_{a=\pi\_{\theta\_{t}}(s\_{i})}\Bigg|\mathscr{F}\_{t}\Bigg]
> $
>
> Now apply Theorem 1, where we note that the result is valid if $\psi\_{t+1}$ is chosen as a function of $\mathscr{F}\_{t},{\cal B}\_{t+1}$, *but independent of the $\boldsymbol{u}\_{i}$'s used in the quasi-gradient construction*, as it exactly happens in the implementation of oCPG (line 13). We obtain
>
> $
> \Bigg\Vert\dfrac{1}{(1-\gamma)N\_{t}}\sum\_{i=1}^{N\_{t}}\nabla\_{\theta}\pi\_{\theta\_{t}}(s\_{i})\mathbb{E}\_{\boldsymbol{u}}\Bigg[\dfrac{Q\_{\psi\_{t+1}}(s\_{i},a+\mu\boldsymbol{u})-Q\_{\psi\_{t+1}}(s\_{i},a)}{\mu}\boldsymbol{u}\Bigg]\Bigg|\_{a=\pi\_{\theta\_{t}}(s\_{i})}-\nabla\_{\beta\_{t+1}}J(\theta\_{t})\Bigg\Vert \le\dfrac{B\sqrt{p}}{1-\gamma}\Bigg[\dfrac{\varepsilon\_{\mu,\psi\_{t+1}}^{\pi\_{\theta\_{t}}}(\beta\_{t+1})}{\mu}+G\mu\Bigg],
> $
>
> where we explicitly highlight the dependence on the (random) PRE $\varepsilon\_{\mu,\psi\_{t+1}}^{\pi\_{\theta\_{t}}}$ on the replay buffer at $t+1$ with $\beta\_{t+1}$, i.e.,
>
> $
> \varepsilon\_{\mu,\psi\_{t+1}}^{\pi\_{\theta\_{t}}}(\beta\_{t+1}) = \sqrt{\dfrac{1}{N\_{t}}\sum\_{i=1}^{N\_{t}}\mathbb{E}\_{\boldsymbol{u}}\big[\big|Q\_{\psi\_{t+1}}(s\_{i},\pi\_{\theta\_{t}}(s\_{i})+\mu\boldsymbol{u})-Q^{\pi\_{\theta\_{t}}}(s\_{i},\pi\_{\theta\_{t}}(s\_{i})+\mu\boldsymbol{u})\big|^{2}\big]}
> $
>
> By the triangle inequality, we can also decompose the PRE as
>
> $
> \varepsilon\_{\mu,\psi\_{t+1}}^{\pi\_{\theta\_{t}}}(\beta\_{t+1}) \le\sqrt{\dfrac{1}{N\_{t}}\sum\_{i=1}^{N\_{t}}\mathbb{E}\_{\boldsymbol{u}}\big[\big|Q\_{\psi\_{t+1}}(s\_{i},\pi\_{\theta\_{t}}(s\_{i})+\mu\boldsymbol{u})-Q'\_{{\cal B}\_{t+1}}(s\_{i},\pi\_{\theta\_{t(i)}}(s\_{i})+\mu\boldsymbol{u})\big|^{2}\big]}
> $
>
> $
> \quad\quad+\sqrt{\dfrac{1}{N\_{t}}\sum\_{i=1}^{N\_{t}}\mathbb{E}\_{\boldsymbol{u}}\big[\big|Q'\_{{\cal B}\_{t+1}}(s\_{i},\pi\_{\theta\_{t(i)}}(s\_{i})+\mu\boldsymbol{u}')-Q^{\pi\_{\theta\_{t}}}(s\_{i},\pi\_{\theta\_{t}}(s\_{i})+\mu\boldsymbol{u})\big|^{2}\big]}
> $
>
> $
> \le L\_{\psi}\sqrt{\dfrac{1}{N\_{t}}\sum\_{i=1}^{N\_{t}}\big|\pi\_{\theta\_{t}}(s\_{i})-\pi\_{\theta\_{t(i)}}(s\_{i})\big|^{2}}
> $
>
> $
> \quad\quad+\sqrt{\dfrac{1}{N\_{t}}\sum\_{i=1}^{N\_{t}}\mathbb{E}\_{\boldsymbol{u}}\big[\big|Q\_{\psi\_{t+1}}(s\_{i},\pi\_{\theta\_{t(i)}}(s\_{i})+\mu\boldsymbol{u})-Q'\_{{\cal B}\_{t+1}}(s\_{i},\pi\_{\theta\_{t(i)}}(s\_{i})+\mu\boldsymbol{u})\big|^{2}\big]}
> $
>
> $
> \quad\quad\quad\quad+\sqrt{\dfrac{1}{N\_{t}}\sum\_{i=1}^{N\_{t}}\mathbb{E}\_{\boldsymbol{u}}\big[\big|Q'\_{{\cal B}\_{t+1}}(s\_{i},\pi\_{\theta\_{t(i)}}(s\_{i})+\mu\boldsymbol{u})-Q^{\pi\_{\theta\_{t}}}(s\_{i},\pi\_{\theta\_{t}}(s\_{i})+\mu\boldsymbol{u})\big|^{2}\big]}
> $
>
> $
> \triangleq\varepsilon'\_{\pi\_{\theta\_{t}}}(\beta\_{t+1})+\varepsilon'\_{\mu,\psi\_{t+1}}(\beta\_{t+1})+\varepsilon'\_{\mu,\pi\_{\theta\_{t}}}(\beta\_{t+1}),
> $
>
> where $Q'\_{{\cal B}\_{t+1}}$ denotes the approximate $Q$-values resulting by using the clipped double $Q$-learning (heuristic) procedure, as in lines $9$ to $10$ of the proposed algorithm. Taking conditional expectations relative to $\mathscr{F}\_{t}$ on both sides and by using Jensen on the left-hand side, we obtain
>
> $
> \big\Vert\mathbb{E}\big[\hat{\nabla}\_{\beta\_{t+1}}^{\mu,\psi\_{t+1}}J(\theta\_{t})\big|\mathscr{F}\_{t}\big]-\mathbb{E}\big[\nabla\_{\beta\_{t+1}}J(\theta\_{t})\big|\mathscr{F}\_{t}\big]\big\Vert
> $
>
> $\le\dfrac{B\sqrt{p}}{1-\gamma}\Bigg[\dfrac{\mathbb{E}\Big[\varepsilon\_{\mu,\psi\_{t+1}}^{\pi\_{\theta\_{t}}}(\beta\_{t+1})\Big|\mathscr{F}\_{t}\Big]}{\mu}+G\mu\Bigg]
> $
>
> $
> \le\dfrac{B\sqrt{p}}{(1-\gamma)\mu}\Bigg[\underbrace{\mathbb{E}\big[\varepsilon'\_{\mu,\psi\_{t+1}}(\beta\_{t+1})\big|\mathscr{F}\_{t}\big]}\_{\text{Compatibility}}+\underbrace{\mathbb{E}\big[\varepsilon'\_{\mu,\pi\_{\theta\_{t}}}(\beta\_{t+1})\big|\mathscr{F}\_{t}\big]}\_{Q\text{-Error (Irreducible)}}+\underbrace{\mathbb{E}\big[\varepsilon'\_{\pi\_{\theta\_{t}}}(\beta\_{t+1})\big|\mathscr{F}\_{t}\big]}\_{\text{Policy Misalignment (Irreducible)}}\Bigg] +\dfrac{GB\sqrt{p}}{1-\gamma}\mu.
> $

---

> > ### Author Response · Authors · 2025-10-29
> > **Convergence Analysis — Part III**
> >
> > Combining with the decomposition of $\hat{\nabla}\_{\beta\_{t+1}}^{\mu,\psi\_{t+1}}J(\theta\_{t})-\nabla J(\theta\_{t})$ we have the final bound
> >
> > $
> > \big\Vert\mathbb{E}\big[\hat{\nabla}\_{\beta\_{t+1}}^{\mu,\psi\_{t+1}}J(\theta\_{t})\big|\mathscr{F}\_{t}\big]-\nabla J(\theta\_{t})\big\Vert
> > $
> >
> > $
> > \le\dfrac{B\sqrt{p}}{(1-\gamma)\mu}\Bigg[\underbrace{\mathbb{E}\big[\varepsilon'\_{\mu,\psi\_{t+1}}(\beta\_{t+1})\big|\mathscr{F}\_{t}\big]}\_{\text{Compatibility}}+\underbrace{\mathbb{E}\big[\varepsilon'\_{\mu,\pi\_{\theta\_{t}}}(\beta\_{t+1})\big|\mathscr{F}\_{t}\big]}\_{Q\text{-Error (Irreducible)}}+\underbrace{\mathbb{E}\big[\varepsilon'\_{\pi\_{\theta\_{t}}}(\beta\_{t+1})\big|\mathscr{F}\_{t}\big]}\_{\text{Policy Misalignment (Irreducible)}}\Bigg]
> > $
> >
> > $
> > +\underbrace{\mathbb{E}\big[\varepsilon\_{rep}'(\beta\_{t+1})\big|\mathscr{F}\_{t}\big]}\_{\text{Weak Replay Error}}+\dfrac{GB\sqrt{p}}{1-\gamma}\mu,
> > $
> >
> > which we are going to use later on. **Observe that at this point $\psi\_{t+1}$ (and thus $Q\_{\psi\_{t+1}})$ can be freely chosen**, as functions of $\mathscr{F}\_{t},{\cal B}\_{t+1}$ at each iteration $t$. We will return to the issue of determining $\psi\_{t+1}$ again towards the end of our analysis.
> >
> > ***Quasigradient Norm Bound:***
> >
> > Assuming that $Q\_{\psi}$ is uniformly $L\_{\psi}$-Lipischitz (this is reasonable in practice as well), we have
> > $
> > (1-\gamma)^{2}\mathbb{E}[\Vert\hat{\nabla}\_{\beta\_{t}}^{\mu,\psi\_{t+1}}J(\theta\_{t})\Vert\_{2}^{2}]
> > $
> >
> > $
> > = \mathbb{E}\Bigg[\Bigg\Vert\dfrac{1}{N}\sum\_{i\in S(t+1)}\nabla\_{\theta}\pi\_{\theta\_{t}}(s\_{i})\dfrac{Q\_{\psi\_{t+1}}(s\_{i},a+\mu\boldsymbol{u}\_{i})-Q\_{\psi\_{t+1}}(s\_{i},a)}{\mu}\boldsymbol{u}\_{i}\Bigg|\_{a=\pi\_{\theta\_{t}}(s\_{i})}\Bigg\Vert^{2}\Bigg]
> > $
> >
> > $
> > \le\mathbb{E}\Bigg[\dfrac{1}{N}\sum\_{i\in S(t+1)}\Bigg\Vert\nabla\_{\theta}\pi\_{\theta\_{t}}(s\_{i})\dfrac{Q\_{\psi\_{t+1}}(s\_{i},a+\mu\boldsymbol{u}\_{i})-Q\_{\psi\_{t+1}}(s\_{i},a)}{\mu}\boldsymbol{u}\_{i}\Bigg|\_{a=\pi\_{\theta\_{t}}(s\_{i})}\Bigg\Vert^{2}\Bigg]
> > $
> >
> > $
> > \le B^{2}\mathbb{E}\Bigg[\dfrac{1}{N}\sum\_{i\in S(t+1)}\Bigg\Vert\dfrac{Q\_{\psi\_{t+1}}(s\_{i},a+\mu\boldsymbol{u}\_{i})-Q\_{\psi\_{t+1}}(s\_{i},a)}{\mu}\boldsymbol{u}\_{i}\Bigg\Vert^{2}\Bigg|\_{a=\pi\_{\theta\_{t}}(s\_{i})}\Bigg]
> > $
> >
> > $
> > \le B^{2}L\_{\psi}^{2}\mathbb{E}\bigg[\dfrac{1}{N}\sum\_{i\in S(t+1)}\Vert\boldsymbol{u}\_{i}\Vert^{4}\bigg]
> > $
> >
> > $
> > = B^{2}L\_{\psi}^{2}\mathbb{E}\bigg[\mathbb{E}\bigg[\dfrac{1}{N}\sum\_{i\in S(t+1)}\Vert\boldsymbol{u}\_{i}\Vert^{4}\bigg|S(t+1)\bigg]\bigg]
> > $
> >
> > $
> > = B^{2}L\_{\psi}^{2}\mathbb{E}\bigg[\dfrac{1}{N}\sum\_{i\in S(t+1)}\mathbb{E}\{\Vert\boldsymbol{u}\_{i}\Vert^{4}\}\bigg]
> > $
> >
> > $
> > \le B^{2}L\_{\psi}^{2}(p+4)^{2},
> > $
> >
> > and therefore,
> >
> > $
> > \mathbb{E}[\Vert\hat{\nabla}\_{\beta\_{t}}^{\mu,\psi\_{t+1}}J(\theta\_{t})\Vert\_{2}^{2}]\le\dfrac{1}{1-\gamma}B^{2}L\_{\psi}^{2}(p+4)^{2}.
> > $
> >
> > ***Putting it altogether:***
> >
> > We now assume that the objective $J$ is both $L\_{J}$-Lipschitz and $G\_{J}$-smooth. These assumptions are standard in the related literature, and in similar analyses of various policy gradient methods. They are also somewhat essential, because they actually make policy gradient methods amenable to analysis. By smoothness, we first have that
> >
> > $
> > J(\theta\_{t+1}) \ge J(\theta\_{t})+(\theta\_{t+1}-\theta\_{t})^{\top}\nabla J(\theta\_{t})-G\_{J}\Vert\theta\_{t+1}-\theta\_{t}\Vert^{2}
> > $
> >
> > $
> > = J(\theta\_{t})+\alpha\nabla J(\theta\_{t})^{\top}\hat{\nabla}\_{\beta\_{t}}^{\mu,\psi\_{t+1}}J(\theta\_{t})-G\_{J}\Vert\hat{\nabla}\_{\beta\_{t}}^{\mu,\psi\_{t+1}}J(\theta\_{t})\Vert^{2}
> > $
> >
> > which implies that
> >
> > $
> > \mathbb{E}[J(\theta\_{t+1})|\mathscr{F}\_{t}]\ge J(\theta\_{t})+\alpha\nabla J(\theta\_{t})^{\top}\mathbb{E}\big[\hat{\nabla}\_{\beta\_{t+1}}^{\mu,\psi\_{t+1}}J(\theta\_{t})\big|\mathscr{F}\_{t}\big]-\alpha^{2}G\_{J}\mathbb{E}[\Vert\hat{\nabla}\_{\beta\_{t}}^{\mu,\psi\_{t+1}}J(\theta\_{t})\Vert^{2}].
> > $
> >
> > Using Cauchy-Schwarz and the triangle inequalities, and by Lipschitzness of $J$, we can then write
> >
> > $
> > \mathbb{E}[J(\theta\_{t+1})|\mathscr{F}\_{t}]
> > $
> >
> >
> > $
> > \ge J(\theta\_{t})+\alpha\nabla J(\theta\_{t})^{\top}\big(\mathbb{E}\big[\hat{\nabla}\_{\beta\_{t+1}}^{\mu,\psi\_{t+1}}J(\theta\_{t})\big|\mathscr{F}\_{t}\big]-\nabla J(\theta\_{t})+\nabla J(\theta\_{t})\big)-\alpha^{2}G\_{J}\mathbb{E}[\Vert\hat{\nabla}\_{\beta\_{t}}^{\mu,\psi\_{t+1}}J(\theta\_{t})\Vert^{2}]
> > $
> >
> >
> > $
> > = J(\theta\_{t})+\alpha\nabla J(\theta\_{t})^{\top}\big(\mathbb{E}\big[\hat{\nabla}\_{\beta\_{t+1}}^{\mu,\psi\_{t+1}}J(\theta\_{t})\big|\mathscr{F}\_{t}\big]-\nabla J(\theta\_{t})\big)+\alpha\Vert\nabla J(\theta\_{t})\Vert^{2}-\alpha^{2}G\_{J}\mathbb{E}[\Vert\hat{\nabla}\_{\beta\_{t}}^{\mu,\psi\_{t+1}}J(\theta\_{t})\Vert^{2}]
> > $

---

> ### Author Response · Authors · 2025-10-29
> **Convergence Analysis — Part IV**
>
> $
> \ge J(\theta\_{t})-\alpha\Vert\nabla J(\theta\_{t})\Vert\cdot\Vert\mathbb{E}\big[\hat{\nabla}\_{\beta\_{t+1}}^{\mu,\psi\_{t+1}}J(\theta\_{t})\big|\mathscr{F}\_{t}\big]-\nabla J(\theta\_{t})\Vert+\alpha\Vert\nabla J(\theta\_{t})\Vert^{2}-\alpha^{2}G\_{J}\mathbb{E}[\Vert\hat{\nabla}\_{\beta\_{t}}^{\mu,\psi\_{t+1}}J(\theta\_{t})\Vert^{2}]
> $
>
> $
> \ge J(\theta\_{t})-\alpha L\_{J}\cdot\Vert\mathbb{E}\big[\hat{\nabla}\_{\beta\_{t+1}}^{\mu,\psi\_{t+1}}J(\theta\_{t})\big|\mathscr{F}\_{t}\big]-\nabla J(\theta\_{t})\Vert+\alpha\Vert\nabla J(\theta\_{t})\Vert^{2}-\alpha^{2}G\_{J}\mathbb{E}[\Vert\hat{\nabla}\_{\beta\_{t}}^{\mu,\psi\_{t+1}}J(\theta\_{t})\Vert^{2}].
> $
>
> Taking expectations, rearranging, averaging over $T$ steps and setting a stepsize $\alpha=1/\sqrt{T}$, we get
>
> $
> \dfrac{1}{T}\sum\_{t=1}^{T}\mathbb{E}[\Vert\nabla J(\theta\_{t})\Vert^{2}] \le\dfrac{J^{*}-J(\theta\_{0})}{\sqrt{T}}+\dfrac{L\_{J}}{T}\sum\_{t=1}^{T}\mathbb{E}\big[\big\Vert\mathbb{E}\big[\hat{\nabla}\_{\beta\_{t+1}}^{\mu,\psi\_{t+1}}J(\theta\_{t})\big|\mathscr{F}\_{t}\big]-\nabla J(\theta\_{t})\big\Vert\big]+\dfrac{G\_{J}B^{2}L\_{\psi}^{2}(p+4)^{2}}{(1-\gamma)\sqrt{T}},
> $
>
> which finally implies that
>
> $
> \dfrac{1}{T}\sum\_{t=1}^{T}\mathbb{E}[\Vert\nabla J(\theta\_{t})\Vert^{2}]
> $
>
> $
> \le\dfrac{J^{*}-J(\theta\_{0})}{\sqrt{T}}+\dfrac{G\_{J}B^{2}L\_{\psi}^{2}(p+4)^{2}}{(1-\gamma)\sqrt{T}}+\dfrac{L\_{J}G\_{Q}B\sqrt{p}}{1-\gamma}\mu
> $
>
> $
> \quad\quad+\dfrac{L\_{J}B\sqrt{p}}{(1-\gamma)\mu}\Bigg[\dfrac{1}{T}\sum\_{t=1}^{T}\underbrace{\mathbb{E}\big[\varepsilon'\_{\mu,\psi\_{t+1}}(\beta\_{t+1})\big]}\_{\text{Compatibility}}+\dfrac{1}{T}\sum\_{t=1}^{T}\underbrace{\mathbb{E}\big[\varepsilon'\_{\mu,\pi\_{\theta\_{t}}}(\beta\_{t+1})\big]}\_{Q\text{-Error (Irreducible)}}+\dfrac{1}{T}\sum\_{t=1}^{T}\underbrace{\mathbb{E}\big[\varepsilon'\_{\pi\_{\theta\_{t}}}(\beta\_{t+1})\big]}\_{\text{Policy Misalignment (Irreducible)}}\Bigg]
> $
>
> $
> \quad\quad\quad\quad+\dfrac{1}{T}\sum\_{t=1}^{T}\underbrace{\mathbb{E}\big[\varepsilon\_{\mu,rep}'(\beta\_{t+1})\big]}\_{\text{Weak Replay Error}}.
> $
>
> This is the final convergence bound, parametric relative to the choice of the $\psi\_{t+1}$'s (measurable relative to $\mathscr{F}\_{t},{\cal B}\_{t+1}$). Every such choice of $\psi\_{t+1}$ induces a critic $Q\_{\psi\_{t+1}}$, which numerically determines the errors above. Further, note that in the above bound, **we have rigorously shown stability of oCPG relative to errors introduced by**
> 1. compatibility,
> 2. $Q$-function approximation,
> 3. and the use of the replay buffer (also inducing policy misalignment).
>
> Importantly, the nonasymptotic behavior of the algorithm depends on the **average values of those errors** over the horizon of operation. Out of those errors, the one which is addresable (and reducable) is indeed the compatibility error, as advocated in our work.
>
> Lastly, note the tension introduced by the presence of $\mu$ both in the numerator and denominator of different terms in the bound. This suggest a sweet spot for $\mu$ in order to make oCPG perform at the best of its ability; this is exactly the behavior we have observed in practice.
>
> Indeed, we may approximate (given our available data in the replay buffer)
>
> $\dfrac{1}{N\_{t}}\sum\_{i=1}^{N\_{t}}\mathbb{E}\_{\boldsymbol{u}}\big[\big|Q\_{\psi\_{t+1}}(s\_{i},\pi\_{\theta\_{t(i)}}(s\_{i})+\mu\boldsymbol{u})-Q'\_{{\cal B}\_{t+1}}(s\_{i},\pi\_{\theta\_{t(i)}}(s\_{i})+\mu\boldsymbol{u})\big|^{2}\big]$
>
> $\approx\dfrac{1}{N}\sum\_{i\in S(t+1)}\big|Q\_{\psi\_{t+1}}(s\_{i},\pi\_{\theta\_{t(i)}}(s\_{i})+\mu\boldsymbol{u}\_{i}^{rep})-Q'\_{{\cal B}\_{t+1}}(s\_{i},\pi\_{\theta\_{t(i)}}(s\_{i})+\mu\boldsymbol{u}\_{i}^{rep})\big|^{2},$
>
> for sufficiently large $N$, by sampling uniformly without replacement from the replay buffer. **Note that this is a statistical regression problem by itself**. This provides justification for determining $\psi\_{t+1}$ as
>
> $\psi\_{t+1}^{oCPG}\in\arg\min\dfrac{1}{N}\sum\_{i\in S(t+1)}\big|Q\_{\psi\_{t+1}}(s\_{i},\pi\_{\theta\_{t(i)}}(s\_{i})+\mu\boldsymbol{u}\_{i}^{rep})-Q'\_{{\cal B}\_{t+1}}(s\_{i},\pi\_{\theta\_{t(i)}}(s\_{i})+\mu\boldsymbol{u}\_{i}^{rep})\big|^{2},$
>
> as it essentially happens in line $11$ of oCPG. $\qquad \square$

---

### Decision · Action_Editor_tRhf · 2025-12-04

**Recommendation:** Reject

**Additional Comments:**

In my explanation for the criterion about the claims, I provided the details for the revision that is requested by the reviewing team.

**Audience:**

Yes

**Audience Explanation:**

This work focuses on actor-critic methods with function approximation which is highly relevant to TMLR's audience due to the importance of actor-critic methods in practical applications.

**Claims And Evidence:**

No

**Claims Explanation:**

All the reviewers acknowledge the contribution of the paper, that is, incorporating a critic such that the Q-function $Q^{\pi_\theta}$ is learned by using function approximation to be used in the policy gradient algorithm. The main idea is to use a Gaussian smoothing idea that is classical in optimization to estimate the action-value gradient by using samples of the $Q$ function and then learning this by using a critic.

However, during the review process, especially thanks to the extremely detailed review of Reviewer pnxs and their follow-up comments, it has became clear that the submission missed some very important and relevant references such as the work of [Kumar et al., 2020] who studied a very closely related policy gradient method by using a similar zero-order estimation, but without the critic. I agree with Reviewer pnxs that comparison with this highly relevant work requires a major revision.

Moreover, the authors introduced a convergence analysis, as a response to Reviewer pnxs comments) during the discussion period which showed that first-order stationarity for the developed method by using similar ideas to [Kumar et al., 2020] but with a critic extension. This is an important addition to the submission. Since Reviewer pnxs mentioned that the extension may be straightforward, it is worth noting that the significance of a theoretical extension cannot be a basis of rejection due to TMLR rules. However, the fact that such a major addition is done during the discussion process means that a new round of reviewing will be necessary for the new results to be verified (hence for the evidence for the claims to be accurate). Moreover, as Reviewer pnxs also stated, the incorporation of the theoretical results to the rest of the paper, by presenting the precise setting, assumptions and, the most importantly, comparison with the literature (such as to the work of [Kumar et al., 2020] and beyond), both in terms of results and in terms of analysis techniques is necessary for the claims in the paper to be supported by accurate, convincing and clear evidence.

As such, my recommendation is for the authors to revise their work by using their new theoretical result and by using Reviewer pnxs suggestions to make the positioning of their contribution in the literature clear. To repeat, the main concern is not the significance of the extension (as clearly stated in TMLR's criteria), however the main concern is the lack of a correct positioning of the result and the fact that a major theoretical result is introduced during the discussion process. Hence, the recommendation of the reviewing team is for the authors to integrate their results in the discussion period into their paper and then to make an extensive, clear and precise comparison with the literature such as the work of [Kumar et al., 2020] and other relevant works pointed out by Reviewer pnxs. Once a precise positioning is done, the authors are welcome to submit a revised version and I will try to assign the new version to the same set of reviewers for a fair evaluation.

[Kumar et al., 2020] Kumar, H., Kalogerias, D. S., Pappas, G. J., & Ribeiro, A. (2020). Zeroth-order deterministic policy gradient. arXiv preprint arXiv:2006.07314.

**Resubmission Of Major Revision:**

The authors may consider submitting a major revision at a later time.